# Interaction of two MADS-box genes leads to growth phenotype divergence of all-flesh type of tomatoes

Baowen Huang [1,2,6], Guojian Hu [1,2,6], Keke Wang[1,2], Pierre Frasse[1,2], Elie Maza [1,2], Anis Djari [1,2], Wei Deng[3], Julien Pirrello[1,2], Vincent Burlat [2], Clara Pons [4], Antonio Granell[4], Zhengguo Li [3,5 ✉], Benoît van der Rest [1,2 ✉] & Mondher Bouzayen [1,2,5 ✉]

All-flesh tomato cultivars are devoid of locular gel and exhibit enhanced firmness and improved postharvest storage. Here, we show that *SlMBP3* is a master regulator of locular tissue in tomato fruit and that a deletion at the gene locus underpins the All-flesh trait. Intriguingly, All-flesh varieties lack the deleterious phenotypes reported previously for *SlMBP3* under-expressing lines and which preclude any potential commercial use. We resolve the causal factor for this phenotypic divergence through the discovery of a natural mutation at the *SlAGL11* locus, a close homolog of *SlMBP3*. Misexpressing *SlMBP3* impairs locular gel formation through massive transcriptomic reprogramming at initial phases of fruit development. *SlMBP3* influences locule gel formation by controlling cell cycle and cell expansion genes, indicating that important components of fruit softening are determined at early pre-ripening stages. Our findings define potential breeding targets for improved texture in tomato and possibly other fleshy fruits.

[1] Université de Toulouse, INRAe/INP Toulouse, UMR990 Génomique et Biotechnologie des Fruits, Avenue de l'Agrobiopole, Castanet-Tolosan F-31326, France. [2] Laboratoire de Recherche en Sciences Végétales - UMR5546, Université de Toulouse, CNRS, UPS, Toulouse, INP, France. [3] Key Laboratory of Plant Hormones and Development Regulation of Chongqing, School of Life Sciences, Chongqing University, 401331 Chongqing, China. [4] Instituto de Biología Molecular y Cellular de Plantas, Consejo Superior de Investigaciones Cientificas- Universidad Politécnica de Valencia, 46022 Valencia, Spain. [5] Center of Plant Functional Genomics, Institute of Advanced Interdisciplinary Studies, Chongqing University, 401331 Chongqing, China. [6] These authors contributed equally: Baowen Huang, Guojian Hu. ✉email: zhengguoli@cqu.edu.cn; benoit.van-der-rest@ensat.fr; bouzayen@ensat.fr

Tomato is a fleshy fruit characterized by an incredible diversity of shape, colour and texture. This diversity, whether of natural or induced origin, offers formidable levers for genetic improvement of these traits to the satisfaction of consumers, processors and producers. Softening and internal fruit structure are traits of prime importance for tomato sensory quality, postharvest behaviour and overall commercial value[1,2]. It is also a major criterion in determining whether the crop is better suited for fresh consumption or the processing industry. The dynamics of fleshy fruit tissue texture is complex, and softness has always been addressed as a ripening associated process[3–5]. Therefore, the control of fruit softening has been mainly achieved, so far, through slowing down the ripening process which leads to loss of sensory quality.

Locular tissue occupies up to 59% of the total fruit area in some tomato cultivars[6] and in this regard it is critical in determining fruit inner tissues structure. The formation of locular tissue starts at early fruit development when cells undergo division and expansion to form parenchymatous tissues that acquire a typical gel texture when large cells with a vast intercellular space undergo a liquefaction process[7–10]. Natural variability of inner tissues structure defines major traits of tomato fruit. In this regard, tomato germplasms offer a wide range of traditional and modern commercial cultivars with large variability in texture due mainly to their contrasted composition in different tissue types, going from gel-like tissue totally or partially filling the locules to locular cavities filled with a non-jelly tissue. The existing tomato cultivars can be categorized in three main types based on their locular tissue structure: (i) the Gel-rich type with locular cavity filled with a liquefied gel-like tissue, (ii) the aubergine-like All-flesh type with locules filled with a non-jelly tissue, and (iii) the pepper-like Puffy type totally devoid of locular tissue in the extreme variants (Supplementary Fig. 1 and Fig. 1a). The All-flesh types are highly sought after by breeders as the most suitable for the processing industry; however, so far, the genetic and molecular basis underlying the differentiation of locular tissue into a jelly type texture remains poorly understood.

In this work, we show that the MADS-box gene *SlMBP3* is a master regulator of locular gel formation and that its down-regulation combined with a natural mutation of *SlAGL11*, a close homolog of *SlMBP3*, affects plant and fruit development in certain genotypes, thus precluding any potential commercial use.

## Results

**Natural diversity of inner tissues structure in tomato fruit is associated with allelic variation within the *SlMBP3* locus.** Several MADS-box transcription factors have been reported to play an active role in fleshy tissues differentiation[11–13], and *STK* and *Shell* class-D MADS-box genes have been mainly associated with seed development[14,15]. More recently, it was reported that downregulation of the tomato class-D *SlMBP3* leads to the conversion of the Gel-containing cultivar Ailsa Craig (AC) into a non-gel All-flesh type. However, this mutation is associated with deleterious phenotypes such as reduced plant growth, a sharp decrease in fruit size and impaired seed development with complete inability to germinate[16]. Strikingly, these phenotypes are absent in the All-flesh commercial cultivars that produce large fruit with seeds exhibiting optimal germination capacity (Fig. 1a). Therefore, the question arises as to whether *SlMBP3* is the causal factor behind the lack of locular gel in the All-flesh cultivars, as inferred from the engineered AC accession. If so, how then to explain the absence of the small fruit and defective seed development in the All-flesh accessions? Exploration and sequencing of the *SlMBP3* locus have revealed a 405 bp deletion in a region located 77 bp upstream of the 5'-UTR in all six All-flesh

accessions (Fig. 1b). This deletion was not found in the Gel-rich cultivars, as confirmed by PCR amplification (Supplementary Fig. 2a, b). No sequence variation was found in the *SlMBP3* coding sequence in the twelve tomato cultivars regardless of the type of locular gel (Supplementary Fig. 2c). In silico search for putative *Cis*-regulatory elements within the 405 bp sequence deleted in the *SlMBP3* promoter of all-flesh cultivars identified several types of conserved *Cis*-elements, including bZIP, C2H2, CPP, GRAS, HD-ZIP, NAC, Nin-Like, WRKY (Supplementary Fig. 2d). These data indicate that this gene is extremely well conserved and that the deleted segment within the promoter represents the only variation at the *SlMBP3* locus and, thereby, might be responsible for the All-flesh trait.

**SlMBP3 is the causal factor determining locular gel tissue differentiation in tomato fruit.** In the tomato, *SlMBP3* and its closest homolog *SlAGL11* belong to the class D MADS-box family and are highly similar (up to 75% similarity within the coding sequence) to *AtSTK*, the unique class D gene in *Arabidopsis thaliana* (Supplementary Fig. 3a, b). In contrast to Arabidopsis where *AtSTK* mutation results in abnormal ovules, seeds and funiculus[17,18], we previously reported that downregulation of *SlAGL11* in the tomato fails to give any fruit phenotype, while its overexpression can mediate the homeotic conversion of sepals into a fleshy organ[19]. To address the putative role of *SlMBP3* in inner tissues differentiation of tomato fruit, we first assessed its expression at the transcript level in twelve tomato accessions displaying contrasted tissue types in the locular cavity. Remarkably, the presence or absence of locular gel in different tomato accessions tightly correlated with *SlMBP3* transcript levels, the lowest levels found in All-flesh cultivars lacking jelly tissue and the highest in the Gel-rich cultivars (Fig. 1c). Given that all six All-flesh accessions analyzed have exactly the same deletion, this sequence variant is likely responsible for the under-expression of *SlMBP3* in these cultivars (Fig. 1b, c). To further assess the link between the All-flesh trait and the expression level of *SlMBP3*, we generated overexpressing lines in three different All-flesh cultivars (Gades, Redsky and TP366) using either the *SlMBP3* CDS driven by its native promoter (*proSlMBP3::SlMBP3*) or by the CaMV-35S promoter (*35S::SlMBP3*). Overexpression of both constructs resulted in the conversion from All-flesh to jelly type locular tissue in the three accessions with, however, the *35S::SlMBP3* leading to high expression level of *SlMBP3* and complete rescue of locular gel phenotype whereas the *proSlMBP3::SlMBP3* resulted in weaker *SlMBP3* expression level and partial recovery of locular gel formation (Fig. 1d). Remarkably, the shift from non-jelly to jelly type locular tissue in All-flesh lines overexpressing *SlMBP3* is associated with a dramatically enhanced fruit softness starting at as early as 10 days post-anthesis (DPA) (Fig. 1d). This strongly supports the hypothesis that the All-flesh traits are associated with a reduced expression of *SlMBP3*.

The functional significance of *SlMBP3* in locular gel formation was then addressed more extensively through the implementation of either genome editing strategies via CRISPR/Cas9 or RNAi approaches. Twelve independent *SlMBP3*-KO mutant lines were obtained, all predicted to result in truncated proteins lacking the M-I-K-C functional domains (Supplementary Fig. 3c and 4a). All these loss-of-function mutants displayed dramatic phenotypes on fruit locular tissue texture recalling the All-flesh cultivars described above (Fig. 1a) with a complete absence of jelly locular tissue (Fig. 2a and Supplementary Fig. 4a). Consistently, the severity of the fruit phenotypes in the *SlMBP3*-RNAi lines strongly correlates with the extent of downregulation (Supplementary Fig. 5a), further emphasizing the correlation between the

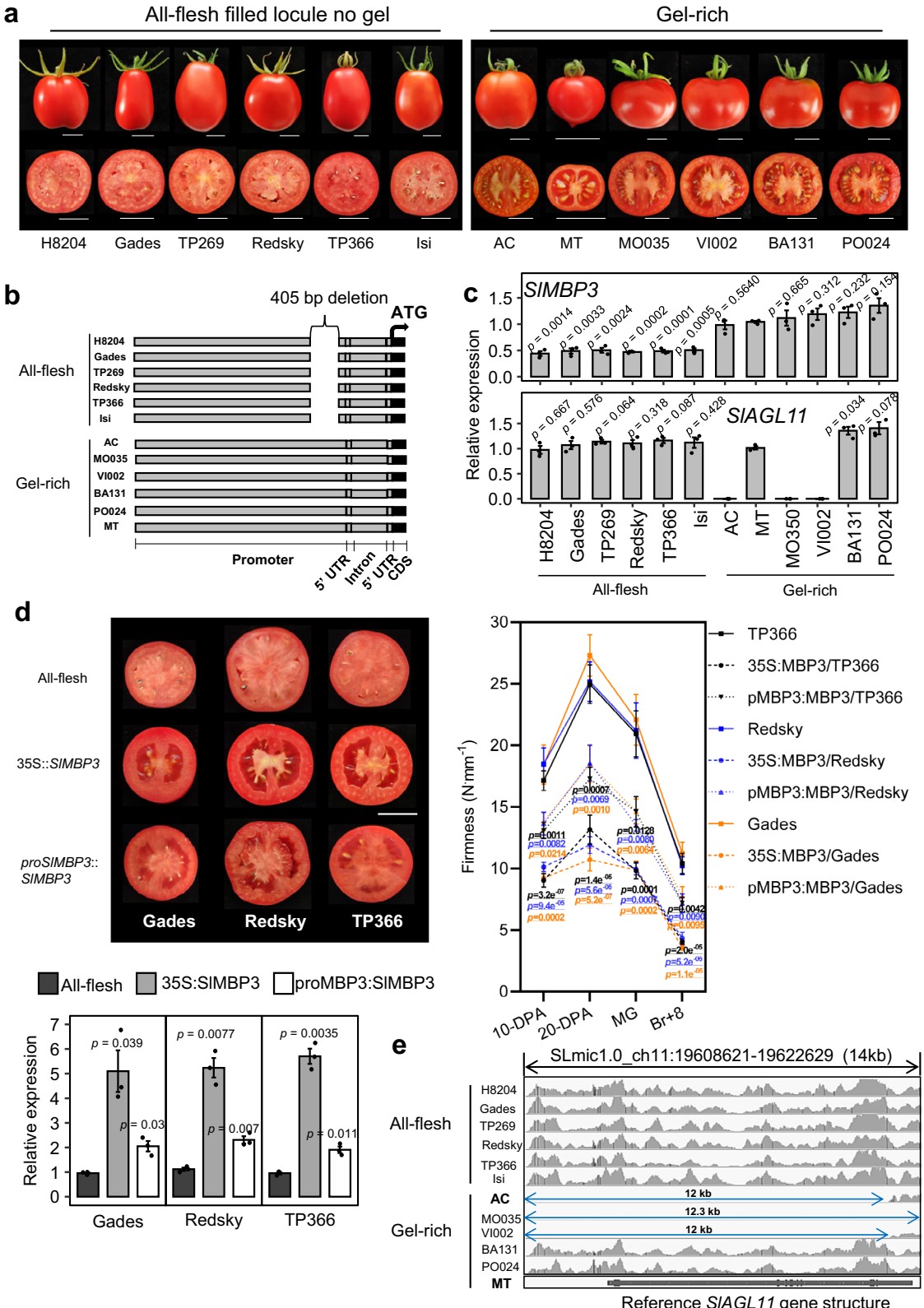

expression level of *SlMBP3* and the formation of locular gel. In this regard, it seems that the jelly *vs* All-flesh trait is determined by a linear dosage effect of *SlMBP3* expression. On the other hand, *SlAGL11* loss-of-function mutations failed to give rise to a distinctive phenotype, whereas double *SlMBP3/SlAGL11*-KO lines showed no locular gel, similar to the *SlMBP3*-KO lines (Fig. 2a and Supplementary Fig. 4b, c). In contrast to the single *SlMBP3*-

KO, dual *SlMBP3/SlAGL11*-KO lines resulted in smaller plants with a dramatic reduction in fruit size and weight and under-developed seeds showing complete inability to germinate (Supplementary Fig. 4d–f).

Interestingly, the reduced fruit size in both dual-KO and dual-RNAi lines recalls the phenotypes of AC tomato cultivars expressing an *SlMBP3*-RNAi construct[16] (Supplementary Figs. 4c

**Fig. 1 Locular tissue development and *SlMBP3* expression levels in various tomato accessions. a** Tomato fruits of cultivars representative of two types of locular tissues defined as All-flesh with no gel and Gel-rich with locules filled with liquefied gel at BR + 8 stage. BR, breaker. **b** Map of structural variants in twelve cultivars. The DNA segment (2.7 kb) upstream of the *SlMBP3* CDS is represented. **c** *SlMBP3* and *SlAGL11* transcript levels were calculated based on the transcript levels of the actin internal reference gene in each line and the outcome of the qPCR data for each line was compared to those of MT to assess change significance using *t*-test in twelve tomato cultivars at 20-DPA fruit stage. DPA, days post-anthesis. Values are means ± standard deviation (SD) of three biological replicates. Significance was determined by two-tailed Student's *t*-test. **d** The left panel shows fruit sections of the All-flesh lines Gades, Redsky and TP366 and their versions expressing either *35S::SlMBP3* or *proSlMBP3::SlMBP3*. The right panel shows the assessment of fruit firmness in All-flesh lines and their versions expressing *35S::SlMBP3* or *proSlMBP3::SlMBP3* at different fruit development stages. Values are means ± SD of *n* = 10 fruits, for each individual line 10 fruits per stage were used and significance was determined by two-tailed Student's *t*-test. All-flesh lines are taken as reference. MG, mature green; BR, breaker. The lower panel shows the *SlMBP3* transcript levels in All-flesh lines and their versions expressing 35 S::*SlMBP3* or *proSlMBP3::SlMBP3* at 20-DPA fruit stage. Values are means ± standard deviation (SD) of three biological replicates. Significance was determined by two-tailed Student's *t*-test. **e** Genomic sequences of *SlAGL11* locus in twelve tomato accessions visualized with Integrative Genomics Viewer showing the deletion (blue arrows) of the complete *SlAGL11* gene in Ailsa Craig (AC), MO035 and VI002 tomato accessions. MicroTom (MT) was used as reference line. Scale bars in (**a** and **d**) represent 2 cm. Source data are provided as a Source Data file.

and 5b). Since our *SlMBP3*-RNAi and CRISPR-KO lines were generated with MT accession and not with AC, we checked whether the phenotypes discrepancy between the two accessions might be due to differences in the genetic background. We, therefore, sequenced the *SlAGL11* locus in 12 tomato cultivars, revealing that three accessions, including AC, have a 12 kb deletion resulting in the complete loss of *SlAGL11* gene body (Fig. 1e). This deletion was confirmed by PCR amplification for AC, MO035 and VI002 accessions, but not for All-flesh, MT, BA131 and PO024 accessions (Supplementary Fig. 2e). Consistently, RT-PCR analysis detected no *SlAGL11* transcripts in AC, MO035 and VI002 (Fig. 1c). These data clearly indicate that dual *SlMBP3/SlAGL11*-KO mutation is required for the reduced fruit size, seed development and plant growth phenotypes, hence providing a clear explanation why *SlMBP3* downregulation or KO mutation leads to severe reduction of fruit size in AC but not in MT. Actually, given the information we provide here, the phenotypes of AC *SlMBP3*-RNAi lines described previously[16] correspond to a double *SlMBP3/SlAGL11* mutation. Therefore, any breeding strategy that aims to modulate the expression of *SlMBP3* has also to consider the allelic variation at the *SlAGL11* locus in order to avoid major detrimental side effects on fruit, seed and plant growth associated with the dual *SlMBP3/SlAGL11* mutation.

**SlMBP3 expression rather than that of SlAGL11 is responsible for locular gel tissue formation.** Toluidine blue staining of tomato fruit sections provided a mean to visualize the differences in locular tissue structure/texture discriminating between All-flesh and Gel-rich tissue types[20]. In WT fruit the locular cavities are filled with a jelly-like tissue showing weak or no staining with toluidine blue, in contrast to *SlMBP3*-KO and RNAi fruit where the heavily stained locular tissue lacks the distinctive jelly character and displays cells with thick and well-defined walls similar to the placenta and central columella tissues (Fig. 2b and Supplementary Fig. 5a). Notably, dual *SlMBP3/SlAGL11*-KO fruit displayed locular tissues similar, in all respects, to single *SlMBP3*-KO fruits, with thicker cell walls strongly stained by toluidine blue (Fig. 2b). Consistently, *SlMBP3* overexpressing fruit exhibited an abundant jelly-like tissue with no toluidine blue staining (Fig. 2a, b). These data indicate that *SlMBP3* is instrumental to the developmental program converting the locular tissue from a parenchymatic to a gel type. Importantly, the modified structure of the locular tissue in *SlMBP3*-KO or dual *SlMBP3/SlAGL11*-KO lines is associated with a marked increase of fruit firmness (Fig. 2c). The dramatic impact of *SlMBP3* expression on firmness is illustrated by the ripe *SlMBP3*-KO fruits exhibiting the same firmness than control fruits at the mature green stage. By contrast, firmness was not altered in *SlAGL11*-KO fruits (Fig. 2c).

These data suggest that only the expression of *SlMBP3*, but not that of *SlAGL11*, is required for locular gel differentiation and further confirms the link between fruit firmness and the All-flesh trait described earlier (Fig. 1d).

In *SlMBP3*-KO fruits, the non-jelly tissue becomes visible at 10-DPA stage just as the distinctive gel tissue begins apparent in wild type fruit (Fig. 2b). Complementation of the *SlMBP3*-KO lines with either the *SlMBP3* CDS driven by its native promoter (*proSlMBP3::SlMBP3*) or by the CaMV-35S promoter (*35S::SlMBP3*) resulted in the recovery of a jelly locular tissue exhibiting weak toluidine blue staining (Supplementary Fig. 6a). Likewise, overexpression of *SlMBP3* in All-flesh accessions (Redsky, TP366 and Gades) led to the recovery of jelly tissue formation, further supporting that high expression of *SlMBP3* is required for locular gel differentiation (Supplementary Fig. 6b). The expression of *SlAGL11* driven by the 35S constitutive promoter in the *SlMBP3*-KO lines also restores a wild type locular gel phenotype (Supplementary Fig. 6c). On the other hand, overexpression of either *SlMBP3* or *SlAGL11* in tomato resulted in the conversion of sepals into carpelloid fleshy organs and in an extreme fruit softening starting at early stages (10-DPA) of fruit development (Fig. 2a, c). Altogether, these data support the idea that fruit firmness can be determined at early pre-ripening stages, and complementation experiments suggest that the two class D tomato proteins may share at least partially some similar functions.

It is worth noting that *SlMBP3*-KO tomato fruits exhibited improved shelf life, as shown by their higher resistance to decay when stored at room temperature for several weeks (Fig. 3a). In addition to the extended shelf life, the fruit also displays significantly higher dry matter and sugar content (Fig. 3b, c). These features indicate that the All-flesh trait gives the fruit additional qualities when they are intended for processing, although the resistance to decay is also a useful fresh market trait.

**SlMBP3 is a master regulator of locular tissue development.** To better decipher the role of *SlMBP3* in the control of inner tissues differentiation in tomato fruit, we generated a series of independent transgenic lines expressing the GUS reporter gene driven either by *SlMBP3* or *SlAGL11* promoters. In situ studies revealed that *SlMBP3* expression starts prior to flower pollination and fruit set, and then becomes highest in the locular tissue as it differentiates into a gel and also in the seeds and funiculus throughout fruit development (Fig. 4a). By contrast, *SlAGL11* expression is undetectable before flower fertilization, then starts at the 4-DPA stage in the seeds but remains undetectable in pericarp, locular gel and funiculus (Fig. 4a). The seed preferential expression of *SlAGL11* is further sustained by the absence of its expression in auxin-induced parthenocarpic fruit that develops in the absence of flower pollination/fertilization

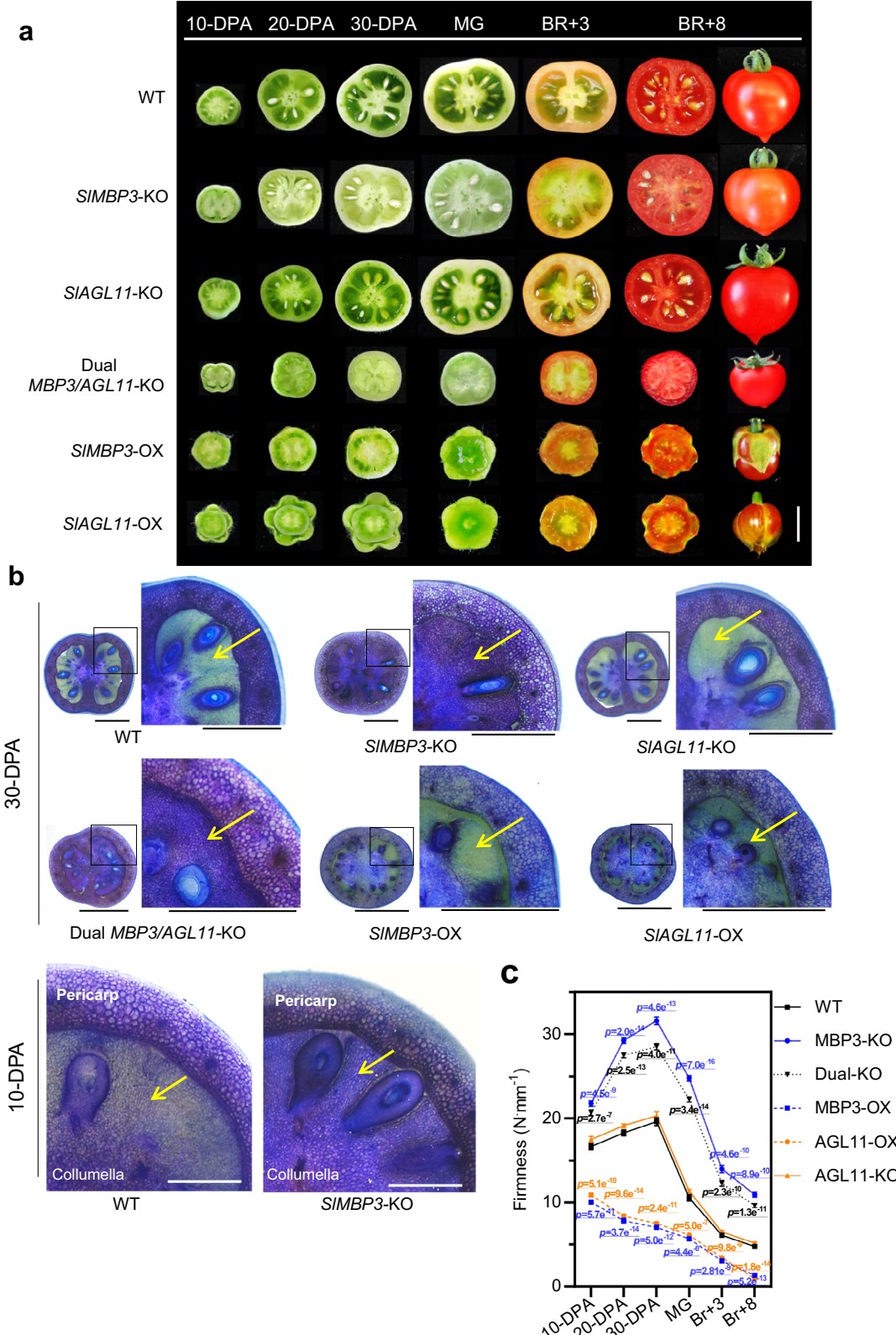

(Fig. 4b). By contrast, *SlMBP3* displayed consistent expression in the placenta of auxin-induced parthenocarpic fruit (Fig. 4b). These distinctive expression patterns are indicative of specific roles for *SlAGL11* and *SlMBP3* in seed development and placental tissue differentiation, respectively.

Global transcriptomic profiling performed by RNA-seq coupled to genome-wide ChIP-seq provided insight into genes and pathways potentially involved in locular tissues formation. *SlMBP3* knock-out leads to substantial transcriptomic reprogramming with 6,807 genes being differentially expressed (DEGs) in *SlMBP3*-KO locular tissue at 10-DPA stage (Fig. 5a and Supplementary Data 1). Among the eight most affected functional categories, the cell wall is the most significant with 46.7% of tomato genes annotated as cell wall-related are DEGs

**Fig. 2 Tomato fruit phenotypes of *SlMBP3* and *SlAGL11* loss-of-function and overexpression lines. a** Cross-sections of tomato fruits of *SlMBP3* and *SlAGL11* knock-out and overexpressing lines throughout fruit development. *SlMBP3*-KO, *SlAGL11*-KO and dual *SlMBP3/SlAGL11*-KO lines were obtained by CRISPR/Cas9-induced mutation as described in Supplementary Fig. 4. Overexpression was achieved by introducing a *SlMBP3* and *SlAGL11* driven by the 35S-CaMV promoter. Scale bar = 1 cm. **b** Histological observations of fruit with altered *SlMBP3* and *SlAGL11* expression. Fruit sections at 30-DPA fruits and 10-DPA fruits were stained with toluidine blue. DPA refers to days post-anthesis. Yellow arrows point to the locular tissue. Black bar = 5 mm and white bar = 1 mm. **c** Assessing firmness of tomato fruit in *SlMBP3* and *SlAGL11* mutant lines. Fruit firmness was monitored at different stages of fruit development and ripening. Values are means ± SD of ten fruits per stage for each individual line, and significance was determined by two-tailed Student's *t*-test. MG, mature green; BR, breaker. Source data are provided as a Source Data file.

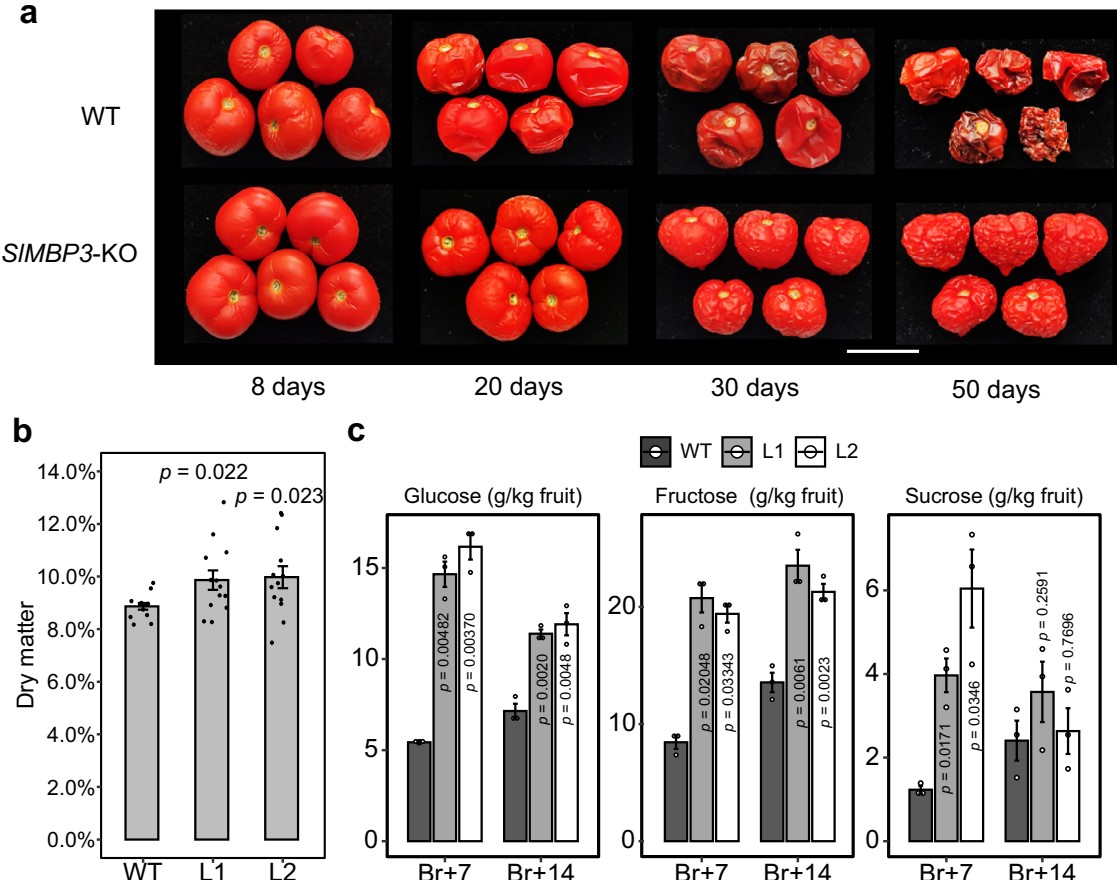

**Fig. 3 Extended shelf life and increased dry matter and sugar contents in *SlMBP3*-KO tomato fruit. a** WT and *SlMBP3*-KO fruit harvested at BR + 7 stage and stored at room temperature. Photos were taken at 8, 20, 30, and 50 days of storage. BR, breaker. **b** Dry matter content expressed as a dry/fresh fruit weight ratio (mean value of 13 fruits at Br+7 stage from each line). Significance was determined by two-tailed Student's *t*-test. **c** Soluble sugar content at two ripening stages assessed by measuring glucose, fructose, and sucrose concentrations. Values are means ± SD of three biological replicates; each replicate includes ten fruits. Significance was determined by two-tailed Student's *t*-test. L1 and L2 are two independent *SlMBP3*-KO lines. Bars = 2 cm. Source data are provided as a Source Data file.

(275/589), and among these, 201 display up-regulation and 74 downregulation in *SlMBP3*-KO locular tissue (Fig. 5b and Supplementary Table 1). Interestingly, global transcriptomic profiling of *SlMBP3*-OX fruit at 10-DPA also revealed that cell wall-related genes are among the top functional categories with 42.4% (250/589) being DEGs, of which 71.2% are downregulated (Supplementary Fig. 7, Supplementary Table 1, and Supplementary Data 2). The altered expression revealed by RNA-seq was confirmed by qRT-PCR analysis for nine cell wall and five transcription factor genes in two independent *SlMBP3*-KO lines (Supplementary Fig. 8). The data suggest that *SlMBP3* plays an active role in controlling the expression of cell wall-related genes at very early stages of locular tissue formation, consistent with the premature and unusual extreme softening of *SlMBP3* overexpressing fruit. *SlMBP3* loss-of-function also has a dramatic

impact on the expression of 524 transcription factor genes from 40 different families (Supplementary Data 3). The high number of DEGs and the many families of transcription factors influenced support the notion that *SlMBP3* behaves as a master regulator of the transcriptomic reprogramming during early fruit development.

A genome-wide ChIP-seq approach was implemented to uncover putative *SlMBP3* target genes using 10-DPA tomato fruits expressing a GFP-tagged *SlMBP3* and displaying phenotypes similar to *SlMBP3*-overexpresing plants, thereby indicating that the chimeric SlMBP3-GFP protein retains functional activity. Sequencing of the immunoprecipitated DNA revealed 2363 peaks (Supplementary Data 4) of which 48.2% were located within a 3-kb region upstream of annotated genes (Fig. 5c). MEME-ChIP software revealed that the most frequent SlMBP3-binding

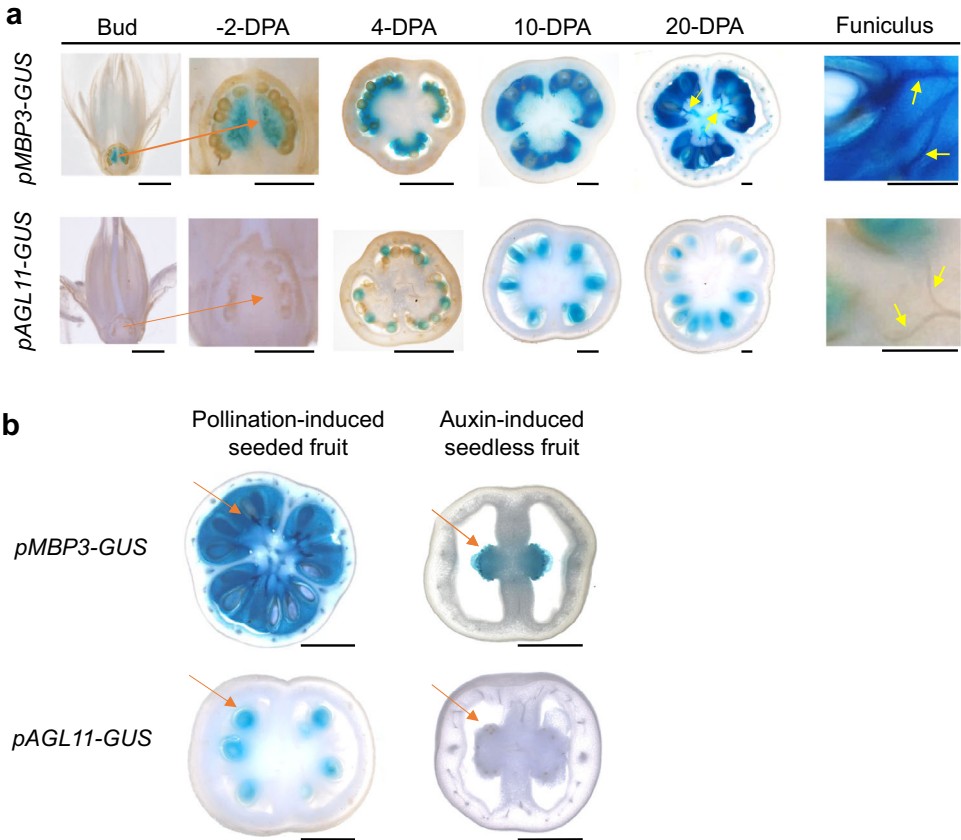

**Fig. 4 SlMBP3 and SlAGL11 expression pattern throughout fruit development. a** *SlMBP3* and *SlAGL11* expression in flower and developing fruit was investigated using transgenic tomato lines expressing the GUS reporter gene driven by the promoter of *SlMBP3* or *SlAGL11*. DPA refers to days post-anthesis; Orange arrows point to the placenta and yellow arrows to funiculus. Scale bar = 1 mm. **b** Expression pattern of *SlAGL11* and *SlMBP3* as visualized by the GUS reporter under the control of *SlAGL11* or *SlMBP3* promoters in pollination-dependent and auxin-induced fruit at 20-DPA stage. Unfertilized flowers were emasculated, and the fruit set was induced either by manual pollination or by exogenous auxin treatment. Orange arrows point to the absence or presence of seeds in parthenocarpic and seeded fruit, respectively. Scale bar = 5 mm.

sequences correspond to a 10-bp CC(A/T)$_6$GG motif of the CArG-box matching the known binding sites of other tomato MADS transcription factors such as RIN, FUL1 and FUL2[21] (Fig. 5d). Taking into account that some genes give rise to more than one peak, the 1139 peaks identified by ChIP-seq correspond to 1081 putative gene targets (Supplementary Data 5). Cross-referencing the ChIP-seq and RNA-seq data identified 450 genes being both *SlMBP3* target and displaying differential expression in *SlMBP3*-KO locular tissue (Fig. 5e and Supplementary Data 6). We, therefore, assumed that cell wall-related genes common to RNA-seq and ChIP-seq data are among the best candidates that may explain the marked firmness exhibited by *SlMBP3*-KO fruits and the extreme softness of *SlMBP3*-OX fruits.

**Cell wall-related genes as key targets of SlMBP3 action**. Among the 450 genes being differentially regulated in *SlMBP3*-KO lines and identified as putative *SlMBP3* targets by ChIP-seq, 24 are related to cell wall, of which 21 out of 24 genes are up–regulated in *SlMBP3*-KO lines and 16 out of 19 are downregulated in *SlMBP3*-OX lines (Supplementary Data 7). These data indicate that the overwhelming majority of cell wall-related genes undergo opposite change in their expression in KO and OX lines (Supplementary Data 7), further supporting their involvement in the dramatic changes of locular tissue texture in *SlMBP3* deficient fruit. Heatmap representation of the expression patterns, extracted from the Tomato Expression Atlas[22], distributed the 24 cell wall-related genes into three distinct clades (Fig. 6a). *SlMBP3* was

included as a reference gene expressed exclusively in locular tissue but not in the pericarp. Clade-I gathers genes displaying high expression in both locular and pericarp tissues, including the putative pectin methylesterase inhibitor gene *SlPMEIL6*, a homolog of Arabidopsis *PMEI6* whose expression has been reported to be under direct regulation of *STK*[14]. Clade-II genes exhibit preferential expression in pericarp where *SlMBP3* shows no expression, suggesting that their repression in the locular gel might be due to a negative regulation by *SlMBP3*. Clade-III genes are expressed in both locular and pericarp tissues, although their expression in locular gel is restricted to the pre-ripening stages. For nine cell wall-related genes, selected as representative of the three clades, ChIP-seq data revealed the presence in the enriched peaks of well-conserved CArG-box motifs known as binding sites for MADS transcription factors (Fig. 6b). Consistent with the outcome of the ChIP-seq experiment, ChIP-qPCR approach performed with tomato fruit expressing GFP-tagged SlMBP3 revealed a substantial enrichment of these nine cell wall-related genes (Fig. 6c). In addition, combined ChIP-seq and RNA-seq analyses identified 61 genes belonging to 16 different transcription factor families (Supplementary Data 8), in line with the substantial transcriptomic reprogramming observed in *SlMBP3*-impaired tomato fruit.

**SlMBP3 influences locular gel formation through regulation of cell expansion and proliferation genes**. Toluidine blue staining used as an efficient mean to discriminate between the liquefied and

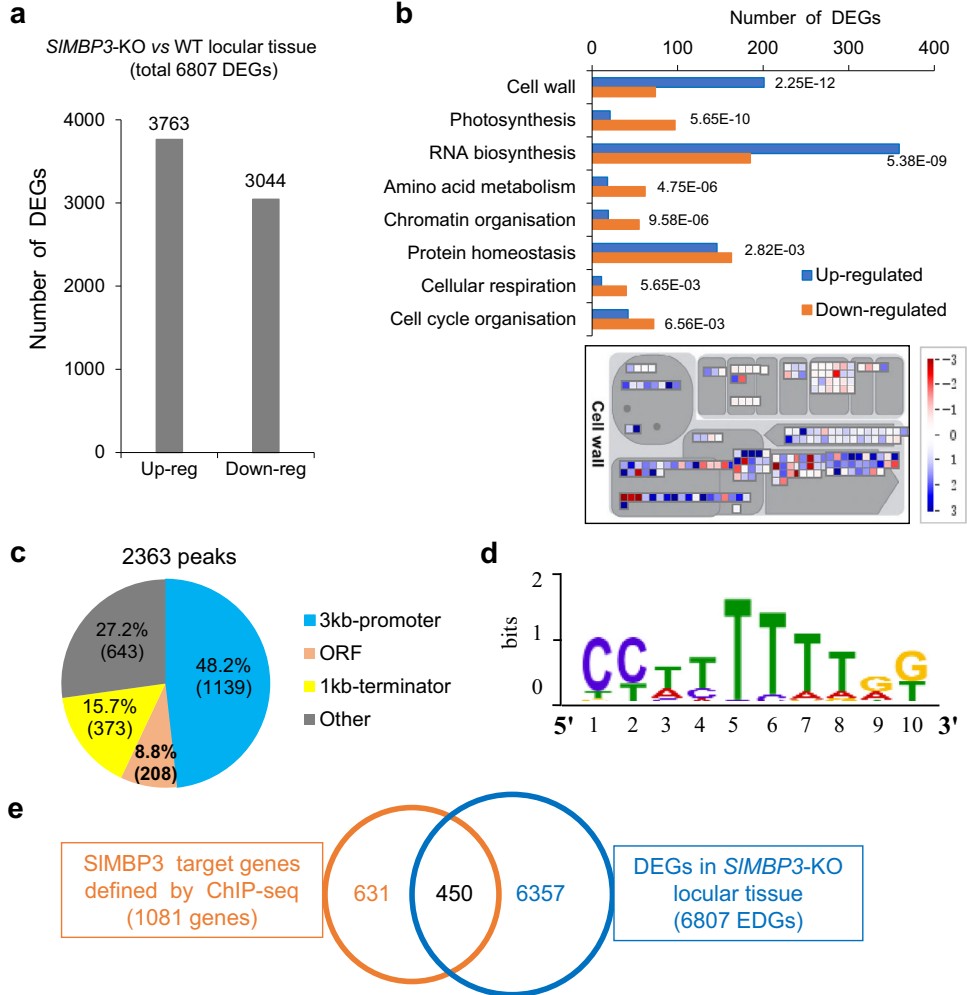

**Fig. 5 *SlMBP3* target genes identified by combined genome-wide RNA-seq and ChIP-seq approaches. a** Differentially expressed genes (DEGs) in *SlMBP3*-KO locular tissue at 10-DPA stage. DPA refers to days post-anthesis. **b** Top eight enriched functional categories in *SlMBP3*-KO locular tissue determined by MAPMAN and showing antagonistic behavior in *SlMBP3* up- and downregulated lines. MAPMAN representation (lower panel) of cell wall-related genes differentially expressed between *SlMBP3*-KO *vs* WT locular tissue. Wilcoxon Rank Sum test was used to evaluate the significance (*p*-value < 0.05) of term enrichment in each category of candidates. DEGs were obtained based on the following rules: basemean > 5 and *padj* < 0.05. **c** Genome-wide distribution analysis of the *SlMBP3*-binding peaks as defined by ChIP-seq analysis at 10-DPA stage. **d** MEME-ChIP analysis of the *SlMBP3*-binding sites within 3-kb-promoters of putative target genes revealing a consensus sequence that matches known CArG-box sequences. **e** Cross-referencing DEGs in *SlMBP3*-KO revealed by RNA-seq and putative direct *SlMBP3* targets identified by ChIP-seq.

non-liquified locular tissues, revealed that the process of locular gel formation is initiated at 8-DPA in WT fruit by contrast to *SlMBP3*-KO fruit where the liquefied tissue never appears in the locules even at later stages (Supplementary Fig. 9). To gain further insight on how SlMBP3 is influencing the differentiation processes of locular tissue resulting in either All-Flesh (*SlMBP3*-KO) or liquefied gel type (WT), we examined the cell types forming the locular tissue in both *SlMBP3*-KO and WT lines. The data show that over-expanded cells start to differentiate in locular tissue of WT fruit at 6 to 9 DPA, concomitant with the appearance of locular gel (Fig. 7). In addition, at 6 and 9-DPA, WT fruit display lower number of cells per mm² than *SlMBP3*-KO lines (Fig. 8a), in line with the higher number of over-expanded cells in WT (Fig. 7). Consistently, the average area of over-expanded cells in WT locular tissue is dramatically bigger than in *SlMBP3*-KO fruit (Fig. 8b) and the cells that appear as disintegrated/fused are not present in *SlMBP3*-KO locular tissue at 9-DPA (Fig. 7). Interestingly, RNA-seq performed on locular tissue at 10-DPA showed that among the genes differentially expressed between WT and *SlMBP3*-KO lines, 114 are annotated as related to cell cycle, cell division and endoreduplication (Supplementary

Data 9). Mapman analysis indicated that cell cycle organization is among the top eight most significantly enriched functional categories (Fig. 5b and Supplementary Data 9). Several cell cycle-related genes being differentially expressed were also identified as putative targets of *SlMBP3* by the ChIP-seq experiment (Supplementary Data 10). Notably, the *CC52A* gene (*Solyc08g080080*), a main marker of endoreduplication[23,24], was downregulated in *SlMBP3*-KO lines while shown to be a putative target of *SlMBP3* (Supplementary Data 10). These data support the hypothesis that *SlMBP3* is driving the formation of liquefied locular tissue through promoting endoreduplication leading to cell expansion and the appearance of disintegrating or fused cells with thin cell walls. It is also noteworthy that a number of auxin-related genes, previously reported to regulate cell division and cell expansion[25,26], were defined as DEGs and as putative direct targets of *SlMBP3* in our study (Supplementary Data 10). Altogether our data clearly indicate that an important contribution to fruit texture and firmness is determined at the very early stages of fruit development, whereas it is commonly accepted that softness is associated with the ripening stage.

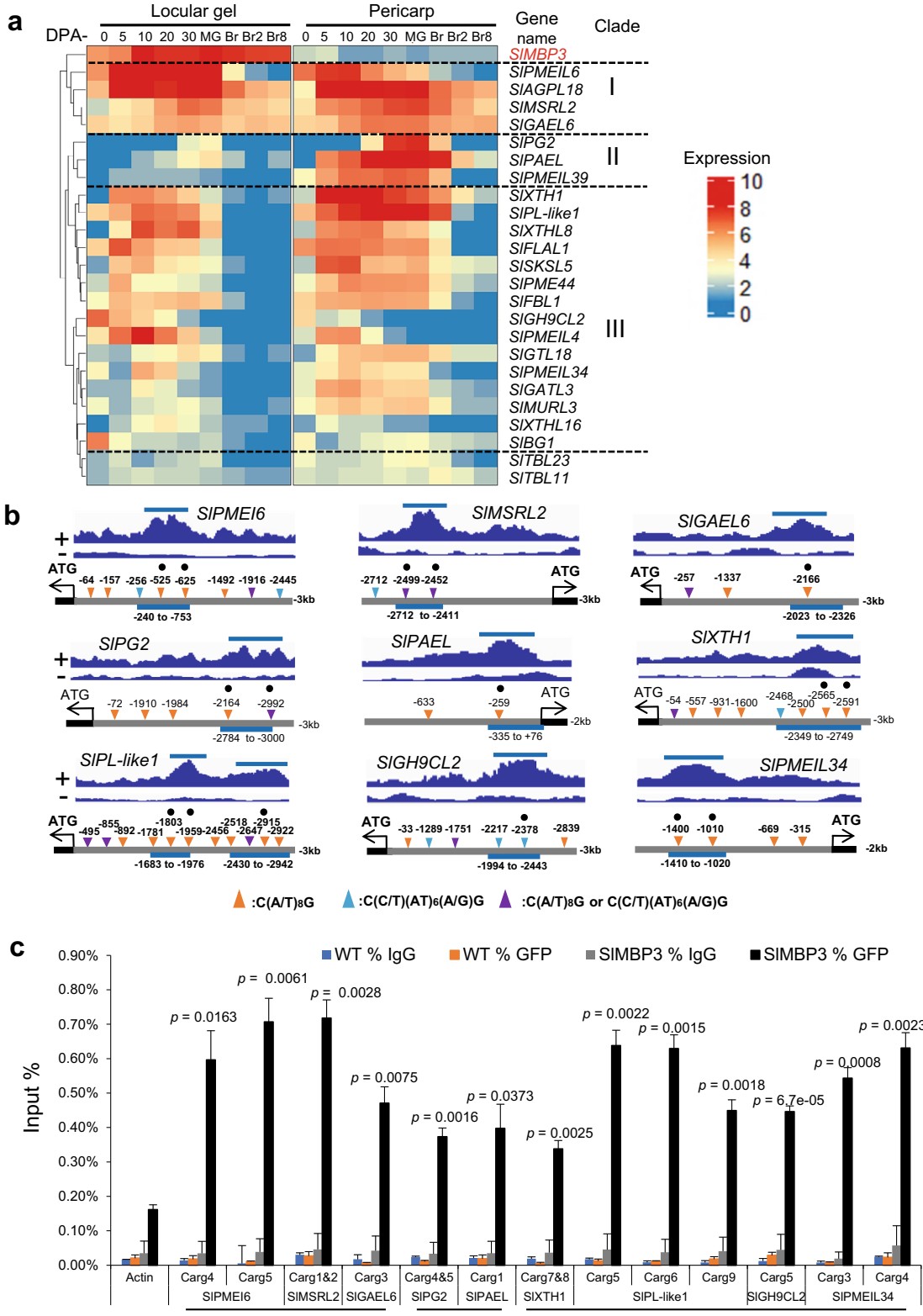

**Fig. 6 Expression pattern of cell wall-related genes differentially expressed in *SlMBP3*-KO lines being putative targets of *SlMBP3*. a** Heatmap representation of the expression patterns of 24 cell wall genes defined as direct *SlMBP3* targets by ChIP-seq and as differentially expressed in locular tissue of CRISPR/Cas9-KO lines. DPA, days post-anthesis; MG, mature green; BR, breaker. **b** DNA enrichment revealed by ChIP-seq experiment and visualized with Integrative Genomics Viewer (up panel) for nine cell wall genes (top panel). + and − correspond to enriched and input samples. For each gene, the localization of three types of CArG-box motifs within a 3-kb promoter region of nine cell wall genes (lower panel) is indicated by triangles. Light blue bars indicate the length of enriched peak. Black circles indicate the position of CArG boxes selected for validation by qPCR based on their high enrichment profile. **c** ChIP-qPCR of nine cell wall genes selected as putative direct targets of *SlMBP3*. Values are means ± standard deviation (SD) of three biological replicates. Significance was determined by two-tailed Student's *t*-test. Source data are provided as a Source Data file.

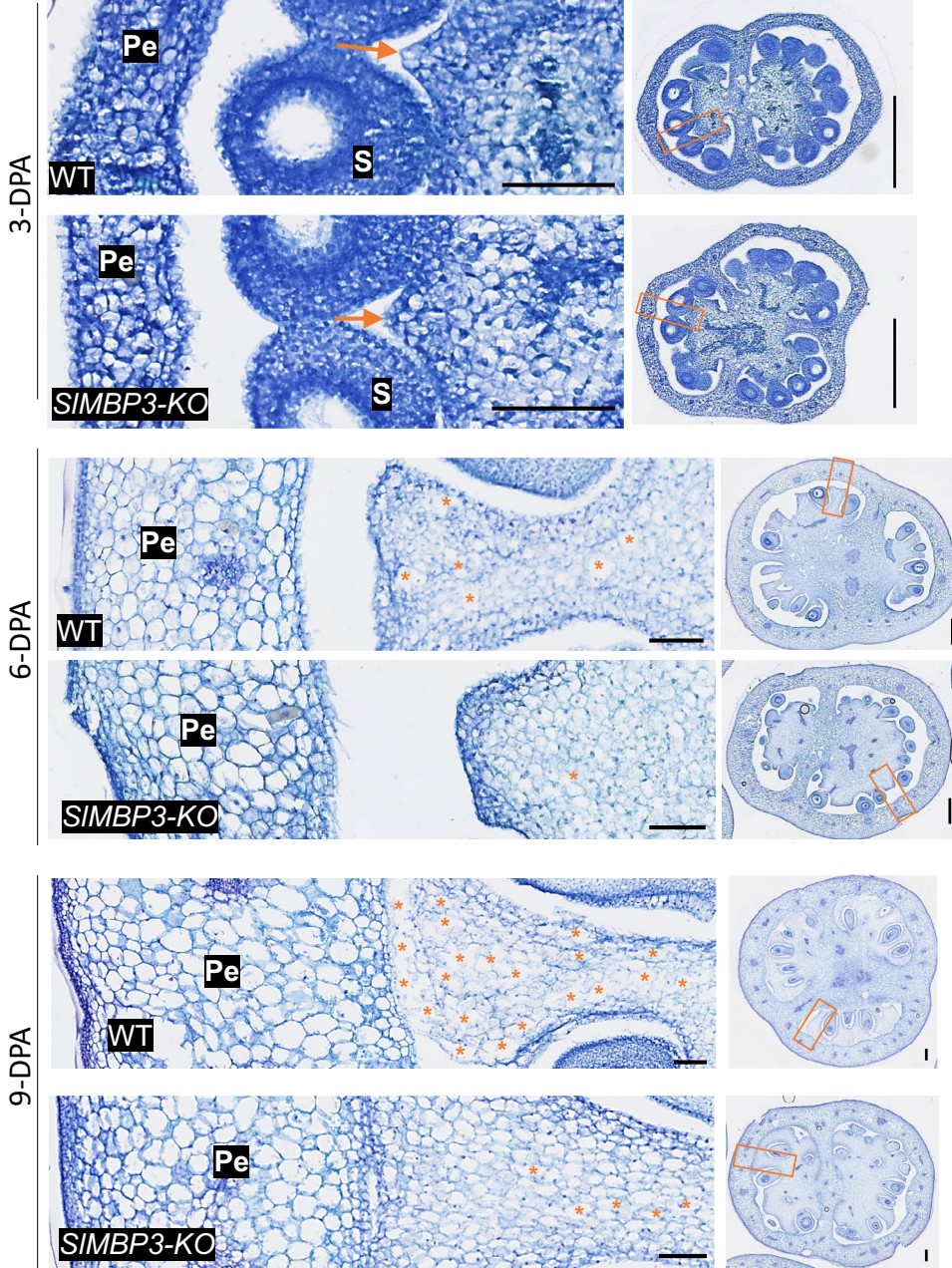

**Fig. 7 Trait variation of the locular tissue cells between *SlMBP3*-KO and wild type fruits.** Paraffin cross-sections of *SlMBP3*-KO and wild type fruit at early fruit development stages (3, 6, and 9 days post-anthesis (DPA)). The orange rectangles on the whole fruit cross-sections on the right are magnified on the left to zoom in on the regions of interest (ROIs) of locular tissue cells used for quantification. Orange arrows indicate the sites of initiation of locular gel formation; * indicate individual over-expanded or disintegrated cells; Scale bar: 100 μm on the left images and 500 mm on the right images; pe pericarp, s seed. Six independent experiments were performed, giving similar results.

## Discussion

The study unravels the role of the class-D MADS-box transcription factor *SlMBP3*, a master regulator of locular tissue in tomato fruit. *SlMBP3*-KO lines are unable to differentiate locular gel tissue, and exhibit a dramatically enhanced firmness, increased dry matter content, and improved postharvest storage. However, it was intriguing that neither the loss-of-function mutation of *SlMBP3* nor the All-flesh commercial tomato varieties were associated with the deleterious phenotypes reported previously in the Ailsa Craig cultivar[16]. The phenotypic discrepancy between the *SlMBP3*-RNAi Ailsa Craig lines and our *SlMBP3*-KO or RNAi lines raised the question as to whether *SlMBP3* is the causal genetic factor behind the absence of locular

gel in the All-flesh cultivars. Actually, the starting point of our study was to explain the absence of small fruits and defective seed development phenotypes in the commercially available All-flesh accessions, as these phenotypes would have precluded their commercial use. Indeed, Ailsa Craig *SlMBP3*-RNAi lines and double *SlMBP3/SlAGL11*-KO MicroTom lines display similar altered vegetative growth with small fruits unable to develop viable seeds. The lack of viable seeds, and thereby progeny lines, likely explains why the previous study was restricted to the characterization of R0 plants[16]. These negative features are absent in our *SlMBP3*-KO lines and available commercial All-flesh varieties that display phenotypes restricted to locular gel while lacking any of the other severe phenotypes. In this regard, a major

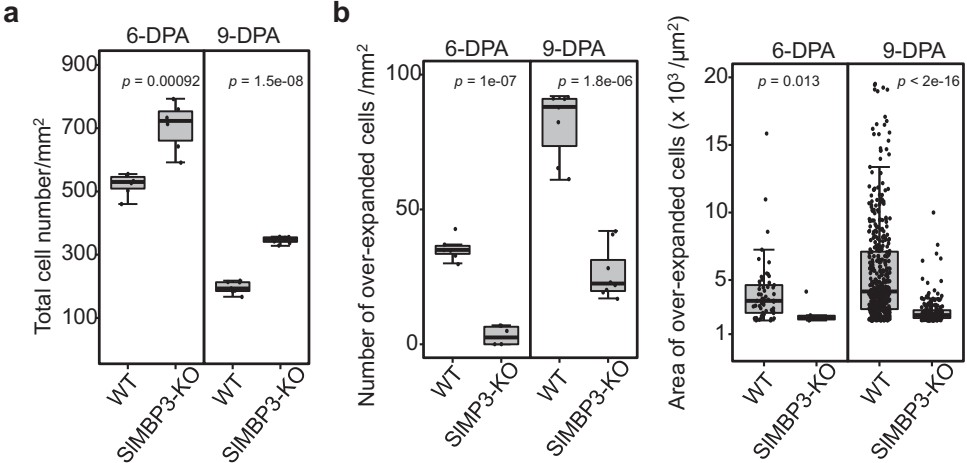

**Fig. 8 Assessing the number and area of cell types in the locular tissue of WT and *SlMBP3-KO* fruit. a** Quantification of the mean total number of cells including over-expanded fused cells per mm² of locular tissue at 6-DPA and 9-DPA. (n = 16–21 ROIs contained in 6–8 fruits per stage coming from six individual plants for each genotype and in the case of *SlMBP3-KO* corresponding to two independent lines). Values are means ± SD, and significance was determined by two-tailed Student's *t*-test. **b** Quantification of the mean number of over-expanded and disintegrated cells per mm² of locular tissue at 6-DPA and 9-DPA (left panel) (n = 18–21 ROIs contained in eight fruits per stage and per genotype). The right panel shows the measurement of the area of these individual cells at 6-DPA and 9-DPA (n > 30 cells per stage and per genotype). Values are means ± SD, and significance was determined by two-tailed Student's *t*-test. Box edges represent the 0.25 and 0.75 quantiles, and the bold lines indicate median values. Whiskers indicate 1.5 times the interquartile range. Note that no measurement was performed at 3-DPA corresponding to the onset of a few locular gel cells in both genotypes. Source data are provided as a Source Data file.

novelty of our finding is to uncover that the causal factor for this phenotypic divergence is a natural mutation at the *SlAGL11* locus, and that this underlies the cryptic genetic variation impacting the phenotypes of *SlMBP3* mutation in different genetic backgrounds. Indeed, combining *SlMBP3* and *SlAGL11* loss-of-functions leads to the deleterious phenotypes erroneously assigned to *SlMBP3* downregulation alone in the previous report[16]. Moreover, we discovered that a deletion in the *SlMBP3* promoter is associated with the All-flesh trait in all commercial tomato varieties. This finding is in line with the outcome of a recent study using a map-based genetic approach that identified a 416-bp deletion in the promoter region of *SlMBP3* is associated with the all-flesh phenotype within the *AFF* tomato cutivars[27]. Nevertheless, this study didn't explain why knocking down *SlMBP3* in certain genotypes leads to deleterious plant and fruit growth and abnormal seed development that hamper any potential commercial use. While our findings provide a lead towards designing tomato varieties with specific inner tissue structure/texture properties, they also show that tomato breeders have inadvertently selected alleles of *SlAGL11*, the *SlMBP3* closest homolog, to avoid major detrimental effects of dual *SlMBP3/SlAGL11* mutants.

Another insight brought by our study is related to the process of determining fruit firmness and texture. Although the deciphering of the components underlying the softening process has been mostly addressed by focusing on late stages of fruit development[3–5], our data are consistent with the notion that a large component of texture and firmness of ripe fruit is determined at early pre-ripening stages, concomitant with the initiation of inner tissues differentiation. In this regard, the outcome of the study brings about a paradigm shift in thinking. Impairing the expression of *SlMBP3* results in massive transcriptomic reprogramming, including a large number of cell wall modifier genes that are under the direct regulation of *SlMBP3*. *SlMBP3*-KO fruits show higher firmness and conversely, the overexpressing fruit exhibit exaggerated softness starting as early as the 10-DPA stage. Therefore, the modulation of fruit texture and inner tissue structure can be achieved via *SlMBP3* leading to extended shelf life and increased dry matter and sugar content without

detrimental effects on fruit size or seed development. Unraveling the molecular events underlying SlMBP3 action by integrating RNA-seq and ChIP-seq data, revealed that impairing *SlMBP3* expression results in massive transcriptomic reprogramming involving several families of transcription factors and a large number of cell wall modifier genes. The high number of DEGs and the many families of transcription factors influenced in the *SlMBP3*-KO lines support the notion that *SlMBP3* is a master regulator of locular tissue differentiation, in line with the increase in its expression occurring concomitantly with the initiation of locular tissue formation. The data reveal the correlation between *SlMBP3* expression and locular gel formation and indicate that locular tissue differentiation into a jelly type requires a high expression level of *SlMBP3*, but a complete silencing of *SlMBP3* is not required to achieve the All-flesh trait as evidenced by the All-flesh accessions or the *SlMBP3*-RNAi lines that retain a significant expression level of this gene. Furthermore, our data show that the formation of locule gel is initiated at very early stages (6–9-DPA), concomitant with the appearance of over-expanded or fused cells in WT fruit that are rarely observed in *SlMBP3*-KO lines. Consistently, RNA-seq identified a number of cell cycle and cell expansion genes that are differentially expressed between WT and *SlMBP3*-KO lines in locular tissue at 10-DPA. Several of these DEGs were defined as a putative direct targets of *SlMBP3* by ChIP-seq, and interestingly, among these some markers of endoreduplication genes were downregulated in All-flesh lines. This supports the notion that *SlMBP3* drives the formation of a liquefied locular gel by promoting endoreduplication and cell expansion leading to the formation of disintegrating or fused cells.

Partial functional redundancy between *SlMBP3* and *SlAGL11* is supported by the defective seed phenotypes observed in dual-KO lines but not in single mutants, suggesting that the presence of one of the two class-D MADS in tomato is sufficient to ensure normal seed development. This hypothesis is also sustained by the ability of *SlAGL11* to complement the *SlMBP3*-KO mutants. Similar functional redundancy has been also reported for class-D MADS in *petunia*[28]. Nevertheless, in tomato the two class-D

MADS seem to be involved in different processes, as evidenced by the distinct phenotypes of the corresponding single KO lines.

Overall, the outcome of the study represents a step forward towards resolving the molecular basis of locular tissue differentiation and, in this regard, provides new opportunities for breeding strategies aiming to design tomato varieties and possibly other fleshy fruits with specific texture properties. Such varieties will be better suited for tomato processing, and the extended shelf life of the fresh fruits will contribute to reducing wastage.

## Methods

**Plant materials and growth conditions**. Tomato (*Solanum lycopersicum* cv MicroTom) and other commercial accessions were grown in culture chambers (14 h day/10 h night cycle, 25/20 °C day/night temperature, 80% relative humidity). After 5% sodium hypochlorite sterilization, tomato seeds were sown on 0,5x Murashige and Skoog medium pH 5.9 with 0.8% (w/v) agar and transferred to the soil after about 16 days. Development and ripening stages refer to days post-anthesis (DPA) and breaker (BR), respectively, as determined by tagging flowers and fruits at anthesis and breaker stages.

**Vector construction and plant transformation**. CRISPR-P (http://cbi.hzau.edu.cn/crispr/) was used to design all the sgRNAs that target either *SlMBP3*, *SlAGL11*, our both genes named here dual *SlMBP3/SlAGL11*. Sequences of the sgRNAs and the induced genomic mutations are given in Supplementary Fig. 4 and Supplementary Data 11. Plasmids were assembled by the Golden Gate strategy[29]. To generate the transgenic plants for RNAi silencing lines, two DNA fragments specific to *SlMBP3* and *SlAGL11* were amplified by PCR from a tomato fruit cDNA library and cloned into pHellsgate12 system vector using the Gateway site-specific recombinational cloning protocols (Invitrogen, USA). For dual *SlMBP3/SlAGL11* silencing, a DNA fragment corresponding to a conserved sequence in *SlMBP3* and *SlAGL11* cDNA was amplified and integrated into the pHellsgate12 system. *SlMBP3* overexpression constructs was obtained after cloning the full length CDS of *SlMBP3* and integration into a PMDC32 vector containing the cauliflower mosaic virus (CaMV) 35S promoter and the Nos terminator using the Gateway® technology (Invitrogen, USA). The 35S::*SlMBP3*-GFP construct was obtained by cloning *SlMBP3* full length sequence into a modified pGreen vector containing the CaMV 35S promoter and the GFP coding sequence downstream of a *SmaI* cloning site[19]. *ProSlMBP3* and *ProSlAGL11* GUS constructs were obtained after amplification of 2,7 kb *SlMBP3* promoter fragment and 1,6 kb *SlAGL11* promoter fragment, respectively, from tomato genomic DNA and insertion into the pMDC162 vector using the Gateway site-specific recombinational cloning protocols (Invitrogen, USA). The p*SlMBP3*::*SlMBP3* construct was built using Golden braid ligation technology (https://gbcloning.upv.es/)[30]. First, domestication steps have been performed as follows, DNA fragments containing the 2,7 kb promoter region of *SlMBP3* and the full length CDS of *SlMBP3* were amplified and cloned into pUPD2 vectors using the *BsmBI* restriction enzyme. In step two, the domesticated p*SlMBP3*, *SlMBP3*, and Nos terminator (https://www.addgene.org, GB0037) part was cloned into pDGBalpha2 vector (*SlMBP3*-alpha2 vector) using *BsaI* restriction enzyme. Then, the final construct, containing the Kanamycin resistance gene, was obtained by mixing the *SlMBP3*-alpha2 vector and Tnos:nptII:Pnos-pDBGalpha1R (https://www.addgene.org, GB0226), into final vector Omega-1 using *BsmBI* enzyme. All detail primers are given in Supplementary Data 11. Transgenic plants were generated by *Agrobacterium*-mediated transformation[19].

**DNA extraction and genotyping of tomato mutants**. For genotyping the first generation (T0) of transgenic line, three different leaf samples from different branches of one plant were collected and total genomic DNA was extracted using the ReliaPrep™ gDNA Tissue Miniprep System (Promega, France). PCR was performed to select the transgenic lines containing the Cas9 construct in T0 generation, and then to select the lines that outcrossed the Cas9 construct in the T1 generation. T2 plants without Cas9 construct were genotyped by PCR using primers designed to amplify a region of 600 bp encompassing the two sgRNA sequences[31]. To further confirm the editing type and check whether the line is homozygous, more than ten plants were grown for each line. PCR-amplified DNA fragments from leaf each plant were cloned into pDonor207 vector using Gateway system, and twelve clones of each PCR product were sequenced to select only the lines with consistent mutation. At least three homozygous lines for each construct were selected for further study.

**Processing data for the WGS of tomato**. Total genomic DNA from young fruits of 12 tomato cultivars (Fig. 1) was extracted using the ReliaPrep™ gDNA Tissue Miniprep System (Promega, France). Library preparation and DNA sequencing using a HiSeq 3000 sequencer (Illumina) was done at Shanghai Majorbio Biopharm Technology Co., Ltd (Shanghai, China), operating in a 150 bp paired-end mode. The raw sequencing reads were first cleaned from their eventual remaining adapters using TrimGalore (a wrapper tool around cutadapt and FastQC,

http://www.bioinformatics.babraham.ac.uk/projects/trim_galore/ and http://www.bioinformatics.babraham.ac.uk/projects/fastqc), then mapped to MicroTom reference genome (http://tomatogenome.gbfwebtools.fr/) using BWA (Burrows–Wheeler Aligner)[32] with default parameters. Only uniquely mapped reads were retained. Following the Broad institute's Best Practices for SNP calling, the sample alignments were then organized by individual subsets called readgroups and cleaned from artifactual duplications caused by the PCR process. A last pre-process step is applied to the data before the SNP calling which consists in the correction of the Base quality scores, performed per-sample, using a machine learning process included in GATK (Genome Analysis Toolkit)[33]. As the base quality score plays an important role in weighing the evidence for or against possible variant, this step tends to correct those scores which correspond to the confidence emitted by the sequencer for each base. The actual variant calling step was managed with HaplotypeCaller from the same toolkit using recommended parameters (https://software.broadinstitute.org/gatk/best-practices/)[33]. The uniquely aligned reads at *SlAGL11* locus were visualized in Integrative Genomic Viewer (IGV) at an interval of 14 kb region of Chromosome 11 (19,608,621-19,622,629). Subsequent genotyping of the *SlMBP3* SNPs and *SlAGL11* locus deletion was done through Sanger sequencing of PCR products. All primer sequences are listed in Supplementary Data 11.

**Phylogenetic tree**. Phylogenetic tree of *Arabidopsis thaliana* and *Solanum lycopersicum* MIKC type MADS-box transcription factors was built after alignment of full-length protein sequences using MUSCLE algorithm and Maximum Likelihood clustering. Analyses were conducted using MEGA6[34].

**Histological observations**. Fruit anatomy was observed using an Axio Zoom V16 microscope (Zeiss, Germany) (FR-AIB TRI imaging platform). Toluidine staining was obtained after dipping hand-cut fruit sections 15 s into 0.05% [w/v] aqueous toluidine blue (Sigma-Aldrich, USA) and water rinsing before mounting the samples under a cover slip.

**Histochemical and statistical analysis**. A higher resolution of the morphological differences of locular tissue cells between *SlMBP3*-KO and WT fruits was analyzed on paraffin sections. Whole fruits from *SlMBP3*-KO and WT at three different early development stages (3, 6, 9-DPA, each 9–15 fruits coming from six individual plants) were harvested by cutting above the crown and rapidly dissected in two halves with a razor blade and immersed in 50 mL Falcon tubes containing FAA fixative solution (10% formalin (37% formaldehyde solution, Sigma-Aldrich, Saint-Quentin Fallavier, France); 50% ethyl alcohol; 5% acetic acid; 35% distilled water). The fixative was infiltrated in the samples by performing ten cycles of 1 min vacuum and vacuum release, and the infiltrated samples were fixed for 16 h at 4 °C. Dehydration series and paraffin infiltration were then performed[16,35]. Large paraffin embedding molds were used to dispose 5–10 fruit halves from both genotypes of a given developmental stage, constituting so-called tissue microarrays. Twelve micrometer-thick serial sections of each tissue array enabled to simultaneously cut under the same conditions large number of fruit cross-sections to be compared. The sections were disposed on silane-coated microscopy slides. Sections were stained for 1 min in 0.05 % Toluidine blue in 0.1 M acetate buffer pH 4.6 for morphology. Following extensive washing, the slides were dried, mounted in Eukitt® (quick-hardening mounting medium, Sigma-Aldrich) and scanned in bright field mode with a 20× objective using a NanoZoomer HT scanner (Hamamatsu, Hamamatsu City, Japan). The scans were observed with NDPview (Hamamatsu, Hamamatsu City, Japan), enabling to extract 16–21 regions of interest (ROIs) containing locular tissue from 6–8 fruits from each developmental stage and genotype observed at 10×. The cells from the locular tissue in these ROIs were analyzed using ImageJ[36].

**GUS staining and assays**. Homozygous T2 transgenic lines expressing the *SlMBP3* promoter (*ProSlMBP3*) or the *SlAGL11* promoter (*ProSlAGL11*) fused to the *GUS* reporter gene were selected based on kanamycin resistance and histochemical staining to confirm GUS activity. Flowers and young fruits samples were incubated at 37 °C in GUS staining solution (0.1% Triton X-Gluc, pH 7.2, 10 mM EDTA[37]; the incubation time ranged from three hours (flowers, very young fruits) to twelve hours (20-DPA fruits). Samples were then depigmented by several washes of graded ethanol series with ordinal 30%, 50%,70%, 95% concentration.

**Auxin-induced fruit set**. Unfertilized flowers were emasculated, and the fruit set was induced by auxin treatment (50 mg/L auxin solution once every 2 days for a period of 7 days).

**Seed germination**. Seed germination was assayed using the following protocol: seeds were gently shaken (50 rpm) in distilled water overnight at 25 °C, and then transferred on moist filter paper in Petri dishes where they were incubated for 7 days at 25 °C. Assays were performed in triplicate (independent biological repeats) with at least 60 seeds.

**Fruit weight**. Fruit weight was estimated by weighing a minimum of 20 fruits harvested at 7 day post-breaker stage (Br + 7) from five different plants bearing the same number of fruit in order not to bias the data.

**Fruit dry matter content**. Dry matter content was estimated using 13 fruits harvested from each line at Br + 7 stage. The weight of each fresh fruit was determined before incubation (Thermosi, SR1000) at 65 °C for 48 h. Then the weight of each dry fruit was measured. Dry matter content was determined by estimating the dry/fresh fruit weight ratio.

**Fruit firmness measurement**. Fruit firmness was assessed using Harpenden calipers (British Indicators Ltd, Burgess Hill, UK)[19]. For each stage, the measures were performed at different development stages on a minimal set of ten fruits for each line.

**Fruit shelf life**. For assessing the shelf life, ten fruits were harvested from different lines at Br+7 stage, pooled and stored at room temperature. Photographs were taken after a storage period of 8, 20, 30, and 50 days.

**Soluble sugar determination**. For each line, ten different plants were used, all bearing the same number of fruit picked at the same ripening stage. Two to three fruits were harvested from each plant and frozen in liquid nitrogen. Glucose, fructose, and sucrose content were determined[19].

**RNA extraction and quantitative RT-PCR**. Fruits from different tomato lines were harvested, frozen in liquid nitrogen, and stored at −80 °C. Total RNA samples were extracted using the RNeasy plant mini kit (Qiagen, Germany). Genomic DNA was removed by DNase treatment (Invitrogen, Cat. No. AM1906). cDNA synthesis was performed using Omniscript Reverse Transcription (Qiagen, Germany). Quantitative real-time PCR (qPCR) was performed in a 10 µL reaction volume using the Takyon PCR Master Mix (Eurogentec, Belgium) on a QS6 sequence detection system (Applied Biosystems, USA). Detailed qPCR primer sequences are listed in Supplementary Data 11.

**RNA-seq analyses and data processing**. Total RNA samples were extracted from tomato locular tissue and whole fruits of 10-DPA stage WT, *SlMBP3*-KO, and *SlMBP3*-OX fruits using Qiagen RNeasy Plant Mini Kit (Qiagen, Germany). For each sample, three independent biological replicates were performed and RNA quality was checked by Agilent 2100 Bioanalyzer to select only samples with rin >8.5. Paired-end RNA sequencing was performed at Shanghai Majorbio Biopharm Technology Co., Ltd (Shanghai, China), using a Truseq RNA Sample Preparation Kit (Illumina) and a Hiseq 2500 platform. The cleaned reads were mapped to the MicroTom reference genome sequence (http://tomatogenome.gbfwebtools.fr/) using HISAT2 2.1.0[38]. HTSeq was then used to calculate raw counts[39]. Differential expression analysis was performed with the DESeq2 R package[40]. Genes were declared as differentially expressed genes (DEGs) if basemean > 5 and adjusted *p*-value (*padj*) < 0.05. DEGs were subjected to the MAPMAN software (version 3.5.1)[41] for functional enrichment analysis. The cell wall-related gene annotation was performed according to www.polebio.lrsv.ups-tlse.fr/ProAnnDB/index.php/.

**ChIP-seq assay and ChIP-qPCR**. Chromatin immunoprecipitation protocol was adapted from ref.[42] with minor modifications. Briefly, tomato fruit at 10-DPA stage expressing a GFP-tagged SlMBP3 protein (SlMBP3-ORF fused with a C-terminal GFP under 35S promoter) were collected from at least six different plants, sliced into small pieces and cross-linked by immersion under vacuum (650–700 mmHg) three times, for 5 min each time, in 1% formaldehyde 1x PBS solution. ChIP assay was performed in three independent biological replicates starting with 2 g of each sample tissue. During nuclei isolation step four washes with extraction buffer II (10 mM Tris-HCl pH 7.4, 0.25 M sucrose, 1% Triton X-100, 10 mM $MgCl_2$, 1 mM PMSF, 50 µM MG132, 1x Roche complete protease inhibitor EDTA free) were needed to reduce chloroplast contamination. Chromatin was isolated by resuspending nuclei pellets in 400 µl of nuclei lysis buffer (20 mM Tris-HCl pH 7.4, 0.5% sarkosyl, 100 mM NaCl, 2 mM EDTA, 1 mM PMSF, 50 µM MG132, 1x Roche complete protease inhibitor EDTA free) and incubation 30 min at 4 °C with gentle agitation (10 rpm). Chromatin was sheared to 300~500 bp using Bioruptor® Pico (BO1060001) with four cycles of 30 s ON and 30 s OFF. Sonicated nuclear lysate was diluted four times with dilution buffer (50 mM Tris-HCl pH 7.4, 1.25% Triton X-100, 100 mM NaCl, 2 mM EDTA, 1 mM PMSF, 50 µM MG132, 1x Roche complete protease inhibitor EDTA free) and 30 µl were collected as Input sample. The remaining lysate was incubated overnight with 2 µL of GFP antibodies (Abcam, AB290, 1:500 dilution) or 5 µL IgG antibodies (Emd Millipore, 12370, 1:200 dilution) at 4 °C, using tube rotator (10 rpm), then coupled 1 h at 4 °C with PBS-BSA washed protein A magnetic beads (Dynabeads SKU10001D). Washing and elution were performed with low salt and high salt buffers[42], and the immunoprecipitated DNA was eluted twice with 100 µl of elution buffer (50 mM Tris-HCl pH 8.0, 10 mM EDTA, 1% SDS). After reverse cross-linked overnight at 65 °C, ChIPed DNA was incubated 30 min at 37 °C with 1 µL RNase cocktail (Ambion, AM2286) then 2h30 at 50 °C with 2 µL Proteinase K

(Ambion, AM2546) to remove RNA and protein contaminations. The Input and ChIP DNA samples were purified using ChIP-DNA clean kit (Zymo Research D5205) and quantified by Qubit® (Qubit dsDNA HS High Sensitivity Assay Kit-Q32851). Ten nanograms of immunoprecipitated DNA was used for library construction and sequencing.

ChIP-sequencing was performed at the GeT-PlaGe core facility (INRAe Toulouse). Sequence libraries were prepared using NEBNext® Ultra™ II DNA Library Prep Kit for Illumina®. Sequencing was performed on an Illumina HiSeq 3000 to produce paired-end reads of 150 bp. Cleaned ChIP-seq reads were aligned to MicroTom reference genome (http://tomatogenome.gbfwebtools.fr/) using BWA (Burrows–Wheeler Aligner)[32] with default parameters, and only uniquely aligned reads were retained. Enriched peak regions in the non-redundant mapped reads were identified by MACS2 v1.4.2 (effective genome size = 770 Mb, *p*-value cutoff = 1.00e-05)[43]. Only the peaks overlapping with a gene in 3-kb upstream region of ATG were considered for further analysis, and the enriched motifs were analyzed by MEME-ChIP software (http://meme-suite.org)[44]. For ChIP-qPCR, the tomato fruit samples were harvested from six different plants and three biological replicates were performed. The primers used for ChIP-qPCR were designed around CArG-box in the enriched regions and are listed in Supplementary Data 11. The fold enrichment was calculated by comparing the Ct values of triplicate measurements between transgenic and wild-type plants.

**Heatmap generation**. Transcriptomic profiling of selected genes was extracted from the Tomato Expression Atlas[22]. The expression pattern in locular gel and pericarp tissue at different development stages was determined for 24 cell wall-related genes shown to be direct targets of *SlMBP3*. The normalized expression values associated with each tissue and stage were extracted from the TomExpress v2020 transcriptomic platform[45]. Heatmap representations were performed with the ComplexHeatmap R package[46]. The distance used for the clustering is based on the Pearson correlation, which clusters genes according to their expression pattern (log2 expression value+1).

**Reporting Summary**. Further information on research design is available in the Nature Research Reporting Summary linked to this article.

## Data availability

All high-throughput sequencing data have been deposited in the European Nucleotide Archive under accession number PRJEB43158. Source data are provided with this paper.

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

## Acknowledgements

The authors are grateful to L. Lemonnier and D. Saint-Martin for transformation and cultivation of tomato plants and GeT-PlaGe core facility (INRAe Toulouse) for ChIP deep sequencing. The authors also want to thank Dr. Christian Chevalier (INRAE et Univsersité de Bordeaux) for helping in analyzing genes related to cell cycle, cell division, and endoreduplication in tomato. This research was supported by the EU H2020 TomGEM 679796 and HARNESSTOM 101000716 projects.

## Author contributions

B.H. performed the experiments and contributed to the drafting of the article; G.H. contributed to the implementation of the experiments and performed the analysis of the RNA-seq and ChIP-seq data; K.W. and W.D. contributed to the generation of the transgenic lines; P.F. contributed to the ChIP experiment, E.M. and A.D. performed the bioinformatic analyses; J.P. contributed to the design of the CRISPR/Cas9 strategies, A.G. and C.P. provided the tomato accessions and contributed to the critical analysis of the results; V.B. contributed to the histological analyses, Z.L., B.v.d.R., and M.B. designed the study, supervised the work, and drafted the manuscript; M.B. conceived and directed the project.

## Competing interests

The authors declare no competing interests.
