## [Peer Review File · Nature Communications]

Interaction of two MADS-box genes leads to growth phenotype divergence of all-flesh type of tomatoesReviewers' Comments:

Reviewer #1:

Remarks to the Author:

This manuscript describes identification of factors defining locular tissue development in tomato fruit. SIMBP3 have been already indicated to be involved in the locular tissue development by an RNAi experiment, but the transformants from the cultivar Ailsa Craig showed several side effects other than defect of locular tissue development, which was not found in commercial "All-flesh" cultivars. The results might have confused the argument of the regulation for locular tissue development. In this study, the authors provide unambiguous evidence that SIMBP3 determines locular tissue development and alleles of SIAGL11, a paralog of SIMBP3, affect the side effects found in the RNAi tomato from Ailsa Craig. The authors found a mutation in SIMBP3 among All-flesh cultivars and a long deletion in the SIAGL11 locus in Ailsa Craig. By CRISPR/Cas9 mediated knockout and RNAi mediated knockdown experiments of either or both SIMBP3 and SIAGL11 indicates partial redundant activities of the two transcription factors but the experiments provided clear-cut evidences that locular tissue development is exclusively dependent on SIMBP3. Over-expression analyses, and other physiological and transcriptome or ChIP analyses strongly support the finding. This study examined almost all possibilities and the results likely support the conclusion perfectly, so I have no doubt or no comment to the conclusion of this manuscript. In this reviewing, the argument for this study may be whether the study provides novelties enough for the standard of this journal. As described by the authors, SIMBP3 has been known to be involved in locular tissue development, although the previous study, the RNAi experiment for Ailsa Craig, did not account for the mechanisms of the all-flesh phenotype found in commercial cultivars. Therefore, it might be easy to focus SIMBP3 as a start point of the study revealing the mechanism for the locular tissue development. However, the experimental designs to find another "cryptic" mutation in Ailsa Craig and determine SIMBP3 as the only factor for the phenotype in the WT background are elegant, and to achieve the simple conclusion, many other possibilities were excluded by developing various kinds of knockout mutants or knockdown/over-expression transformants. In addition, the finding will contribute practical breeding programs for processing tomato cultivars. Therefore, I believe this manuscript has enough originality and novelty. This is just a comment for the authors, if SIAGL11 with a constitutive promoter is overexpressed in all-flesh cultivars or a SIMBP3-KO line, is locular tissue development restored by complementing the SIMBP3 deficiency? The possibility is described in line 4 of page 9 but I could not find the result in this manuscript.

Reviewer #2:

Remarks to the Author:

The manuscript by Huang et al., entitled "Cryptic variation of two MADS-box genes resolves the causal factor of phenotypic variation of locular tissue in tomato fruit thus defining optimal breeding strategies" provides a clear association between the "All flesh" phenotype in tomato and a 405bp deletion in the promoter region of a class-D MADS-box gene, SIMBP3, causing downregulation of the gene. Analysis of the fruit phenotype shown by the SIMBP3-RNAi lines with different degree of silencing also demonstrates that the expression level of the gene correlates with the severity of the "All flesh" trait. The "All-flesh" trait obtained by SIMBP3 knock-out/silencing confers to the fruits enhanced firmness and longer shelf life.

The manuscript also provides evidence that overexpressing SIMBP3 can convert "All Flesh" fruits to jelly type fruits. In addition, it associates reduction in fruit size and seed malformation observed when SIMBP3 is silenced in the cultivar Ailsa Craig with the knock-out of its closest homolog SIAGL11, due to a deletion in the gene sequence. Application of RNA-seq analysis to locular tissue and fruits of SIMBP3 -KO plants coupled with ChIP-seq, allows the identification of differentially expressed genes that are putatively regulated by SIMBP3 at the transcriptional level. These analyses reveal that genes coding for enzymes involved in cell wall modification are key targets of SIMBP3.

This work provides a robust demonstration that SIMBP3 is a master regulator of locular tissue

development in tomato and thus a powerful tool for tomato breeding. However, these results are only partially novel. Clear indications of the role of SIMBP3 in locular gel tissue formation were presented in the work of Zhang et al, (J Exp Bot 2019) along with the observation of the lack of SIAGL11 expression in the cultivar Ailsa Craig. Several pieces of information regarding the expression pattern of SIMBP3 and SIAGL11, the phenotypic traits of SIAGL11-RNAi e SIAGL11-OE plants, and the impact of SIMBP3 and SIAGL11 on the regulation of cell wall- related genes seems to largely confirm previous observations (Zhang et al., 2019, Huang et al. J Exp Bot 2017).

Furthermore, the manuscript only partially elucidates the specific contribution of SIMBP3 and its closest homolog SIAGL11 and their possible interaction in controlling plant and fruit size, and seed development; for example, the discrepancy between the normal vegetative and fruit phenotype of SIAGL11 silenced plants (Huang et al., 2017) and the reduced vegetative growth and fruit size of the double SIMBP3-SIAGL11-KO in the MicroTom background is not discussed. Their respective role in fruit firmness should also be commented upon considering that overexpression of both genes results in an early decrease in firmness.

Genome-wide transcriptomic and ChIP analyses that are well conducted and coupled, represent a powerful method to obtain information about the regulatory activity of SIMBP3 in the process of locular tissue formation. Using this approach, 450 genes were identified that represent putative targets of SIMBP3 and show differential expression in SIMBP3-KO locular tissue. However, the authors focused mainly on cell-wall related genes that represent only a minor portion of the whole set. Since the effects on transcription of gene involved in cell wall metabolism has already been observed (Zhang et al., 2019) in SIMBP3 silenced AC plants, I think it would have been interesting to extend an in-depth analysis to other categories of genes as well.

Overall, I think that although this work provides a lot of information obtained from well-planned and well-conducted experiments, the data presented offer limited novel findings that only partially address the molecular mechanisms underlying the action of SIMBP3 on locular tissue development.

Specific points

Because the 405- bp deletion in the upstream region of the 5' UTR of SIMBP3 causes a reduction in the transcript level of the gene, it would be interesting to analyze this region for the presence of regulatory element.

To confirm the finding that the alterations in fruit size and seeds observed in the SIMBP3 silenced AC lines are directly related to the lack of the SIAGL11 gene, it would be appropriate to downregulate SIMBP3 in a different "Gel-Rich" accession defective for SIAGL11 expression (es MO035).

Complementation experiments performed with overexpression of SIMBP3 and SIAGL11 in *Arabidopsis thaliana* do not seem of much interest for the main objective of this work.

Reviewer #3:

Remarks to the Author:

Huang et al present new information on a gene, MBP3, centrally important in tomato locule formation. They affirm gene function through altered expression and knock out of the MBP3 gene and characterization of a natural promoter mutation that reduces expression. They further demonstrate that MBP3 influences softening through effects that initiate well before ripening begins. Finally, they clarify that seed development and vegetative phenotypes reported in a prior functional characterization of MBP3 were performed in a genotype carrying a natural mutation in the closely related SIAGL11 gene and result from knock out of both genes. This data is important in furthering our understanding of tomato locule development and provides novel insights into determinants influencing softening. A number of issues should be clarified.

The introduction briefly describes the formation of locule tissue as parenchymatous cells that have been described as giant cells by others. The cells forming the eventual locule stop dividing early in development and expand considerably. What was not described is how MBP3 is influencing this process. Are the cells in wild or MBP3 KO lines giant cells that fail to liquify or are they parenchyma

cells more similar to the pericarp? The microscopy suggests they are more like the pericarp but not exactly the same. A small fruited genotype is used while prior descriptions of locule development may have been describing larger fruit. Does the expansion of the locule cells occur in this genotype? The manuscript purports to describe the role of MBP3 in locule formation but actually provides little insight into the cellular changes that result from altered MBP3 expression. Is the process of cellular proliferation and expansion altered or is the role only in conferring cellular liquefaction? With little effort some images of the larger fruited aff genotypes could help clarify what role if any MBP3 is playing beyond cellular liquefaction.

In the manuscript introduction, last paragraph, it is noted that All-flesh" types are highly sought after by breeders for processing and in fresh processed products. The relevance to practical use is cited in the abstract and discussion as well. Is this actually true for fresh use varieties? At least the fresh varieties I see in the market do not seem to carry this trait nor do I see it in the fresh processed products with which I am familiar. aff genotypes are available and the mutation can be readily bred and is definitely in some processing varieties. Quality attributes and consumer preferences for tomatoes are also influenced by juiciness and the locule was long ago shown to be a reservoir of flavor chemistry in addition to the pulp (pericarp). It would seem aff may be less relevant to fresh types. Clarification in this regard for both processing and fresh types would be useful to clarify this point of practical relevance.

Figure 1 d. Complementation of the aff lines with MBP3 under its endogenous promoter provides partial complementation on all genotypes tested while 35S results in what appears to be a normal liquid locule. Could this indicate that all needed regulatory sequences are not present in the promoter sequences used?

It is stated that softening occurs as early as 10 d. Is it more accurate to note that it could be influenced even earlier but 10 d was the first time point examined? Or were there earlier time points with no change tested, and if so such data should be included. This comes up in reference to OE lines and softening at 10 days as well. I imagine looking at earlier stages is perhaps difficult but can anything be said about when the softening differential due to MBP3 can be first identified in normally expressing or OX fruit? If not the text should be modified accordingly.

Expression of MBP3 is presented using relative terms with aff fruit used as reference. This gives minimal indication of the effect on the deletion on MBP3 expression. Is MBP3 still expressed at any appreciable level in aff fruit as a result of the deletion? In the discussion it is stated that aff lines retain significant expression of MBP3 but one never gets a clear picture of what this significant level is. Based on Fig 2c it appears that is about half of WT but one gets no sense of WT expression levels. Perhaps they could provide insight from TomExpress. Also, is the expression in Fig 2c from locule tissue, whole fruit? The figure does not indicate the tissue used and the legend does not clarify.

Line 132 the adjective "subtle" is used describing dosage effect. Is the right terminology? Given the gradation of phenotypes shown in Sup fig 5a perhaps "linear" or "parallel" are more appropriate descriptors?

Lines 176-178. This text is unclear to me. Are they saying that the locule becomes jelly at the 10 DPA stage in WT of the fruit shown in Fig 2 a? The images appear to indicate this is not occurring until 30 DPA. Or is the intent to indicate that the tissue that becomes the liquified gel can be differentiated at this stage? Fig 2 b does indicate lighter staining of WT at 10 DPA but this appears to be much less than at 30 DPA and at least by appearances similar to some of the staining in the supplemental figures where the locules are presented as more solid. In short, the baseline of locule change in the reference genotype should be better described as it is the basis of many of the observations presented.

Figure 2a shows some interesting phenotypes not mentioned in the manuscript. AGL11-KO are clearly liquifying their locules early at 20 DPA where this is not apparent in the WT until 30 DPA. Could AGL11

be providing a repressive effect? In which case MBP3 is still the primary regulator as described but AGL11 may also have a role?

MBP3 and AGL11 OX are presented as having similar phenotypes but it appears the ATL11 OX has seeds in the BR+3 fruit though not in other stages. It appears that the pericarp tissue in 10-30 DPA OX fruit has also become liquified in MG and later stages. Is that true? The dual and OX images are difficult to see. While difficult to describe there appear to be differences between MBP3 and AGL11 OX fruit with OX MBP3 presenting red, light yellow and deep yellow layers (outside to inside) while AGL11 OX presents just two layers, deep red and deep yellow. In this regard, images as in b for stages later than 30DPA might be helpful as would larger and clearer images as in Sup Fig 5.

Why is it stated in lines 189-191 that the two D class genes may have similar function when the KO lines point out they do not? It is stated explicitly in the discussion that they have different functions. This text should be clarified.

Response to Reviewers

The following are our point-by-point answers to the reviewers' questions and comments.

Reviewer #1:

This manuscript describes identification of factors defining locular tissue development in tomato fruit. SIMBP3 have been already indicated to be involved in the locular tissue development by an RNAi experiment, but the transformants from the cultivar Ailsa Craig showed several side effects other than defect of locular tissue development, which was not found in commercial “All-flesh” cultivars. The results might have confused the argument of the regulation for locular tissue development. In this study, the authors provide unambiguous evidence that SIMBP3 determines locular tissue development and alleles of SIAGL11, a paralog of SIMBP3, affect the side effects found in the RNAi tomato from Ailsa Craig. The authors found a mutation in SIMBP3 among All-flesh cultivars and a long deletion in the SIAGL11 locus in Ailsa Craig. By CRISPR/Cas9 mediated knockout and RNAi mediated knockdown experiments of either or both SIMBP3 and SIAGL11 indicates partial redundant activities of the two transcription factors but the experiments provided clear-cut evidences that locular tissue development is exclusively dependent on SIMBP3. Over-expression analyses, and other physiological and transcriptome or ChIP analyses strongly support the finding. This study examined almost all possibilities and the results likely support the conclusion perfectly, so I have no doubt or no comment to the conclusion of this manuscript. In this reviewing, the argument for this study may be whether the study provides novelties enough for the standard of this journal. As described by the authors, SIMBP3 has been known to be involved in locular tissue development, although the previous study, the RNAi experiment for Ailsa Craig, did not account for the mechanisms of the all-flesh phenotype found in commercial cultivars. Therefore, it might be easy to focus SIMBP3 as a start point of the study revealing the mechanism for the locular tissue development. However, the experimental designs to find another “cryptic” mutation in Ailsa Craig and determine SIMBP3 as the only factor for the phenotype in the WT background are elegant, and to achieve the simple conclusion, many other possibilities were excluded by developing various kinds of knockout mutants or knockdown/over-expression transformants. In addition, the finding will contribute practical breeding programs for processing tomato cultivars. Therefore, I believe this manuscript has enough originality and novelty. This is just a comment for the authors, if SIAGL11 with a constitutive promoter is overexpressed in all-flesh cultivars or a SIMBP3-KO line, is locular tissue development restored by complementing the SIMBP3 deficiency? The possibility is described in line 4 of page 9 but I could not find the result in this manuscript.

Answer: The reviewer is right. We have now added this data as Supplementary Fig 6c, showing that the expression of *SIAGL11* driven by a constitutive promoter in the *SIMBP3*-KO lines restores wild type-like locular gel phenotype.

Reviewer #2:

The manuscript by Huang et al., entitled “Cryptic variation of two MADS-box genes resolves the causal factor of phenotypic variation of locular tissue in tomato fruit thus defining optimal breeding strategies” provides a clear association between the “All flesh” phenotype in tomato and a 405bp deletion in the promoter region of a class-D MADS-box gene, *SIMBP3*, causing downregulation of the gene. Analysis of the fruit phenotype shown by the *SIMBP3*-RNAi lines with different degree of silencing also demonstrates that the expression level of the gene correlates with the severity of the “All flesh” trait. The “All-flesh” trait obtained by *SIMBP3* knock-out/silencing confers to the fruits enhanced firmness and longer shelf life.

The manuscript also provides evidence that overexpressing *SIMBP3* can convert “All Flesh” fruits to jelly type fruits. In addition, it associates reduction in fruit size and seed malformation observed when *SIMBP3* is silenced in the cultivar Ailsa Craig with the knock-out of its closest homolog *SIAGL11*, due to a deletion in the gene sequence. Application of RNA-seq analysis to locular tissue and fruits of *SIMBP3* -KO plants coupled with ChIP-seq, allows the identification of differentially expressed genes that are putatively regulated by *SIMBP3* at the transcriptional level. These analyses reveal that genes coding for enzymes involved in cell wall modification are key targets of *SIMBP3*.

- This work provides a robust demonstration that *SIMBP3* is a master regulator of locular tissue development in tomato and thus a powerful tool for tomato breeding. However, these results are only partially novel. Clear indications of the role of *SIMBP3* in locular gel tissue formation were presented in the work of Zhang et al, (J Exp Bot 2019) along with the observation of the lack of *SIAGL11* expression in the cultivar Ailsa Craig. Several pieces of information regarding the expression pattern of *SIMBP3* and *SIAGL11*, the phenotypic traits of *SIAGL11*-RNAi e *SIAGL11*-OE plants, and the impact of *SIMBP3* and *SIAGL11* on the regulation of cell wall- related genes seems to largely confirm previous observations (Zhang et al., 2019, Huang et al. J Exp Bot 2017).

Answer: We respectfully disagree with the reviewer here. One major novelty of our finding is to uncover that the severe detrimental phenotypes on plant growth, fruit size and seed formation are due to dual mutation of both *SIMBP3* and *SIAGL11* which has never been reported before. These vegetative and reproductive growth phenotypes are not only due to *SIMBP3* as reported in Zhang et al. 2019. Indeed, the complete knock-out of

SIMBP3 results in detrimental effects on vegetative and reproductive growth only in a genetic background defective in *SIAGL11*. Our data reveal for the first time that several tomato genotypes like Ailsa Craig have a big deletion (12 kb) at the *SIAGL11* locus. This is the causal factor for the phenotypic divergence between the phenotypes we describe for *SIMBP3*-silenced lines and those reported by Zhang et al. 2019. This natural mutation at the *SIAGL11* locus underlies the cryptic genetic variation impacting the phenotypes of *SIMBP3* mutation in different genetic backgrounds. We think this information is instrumental for designing efficient breeding strategies aiming to gain the All-flesh trait.

A second major novel insight brought by our study is related to the process determining fruit firmness and texture. So far, the deciphering of the components underlying the softening process has been mostly addressed by focusing on late stages of fruit development, namely the ripening phase, however, our data support the notion that a large component of texture and firmness of ripe fruit is determined at early pre-ripening stages, concomitant with the initiation of inner tissues differentiation.

Another novel insight provided by our data is the identification of a number of genes that are direct targets of *SIMBP3*, including cell-wall, cell division and endoreduplication-related genes as well as a number of transcriptional regulators that are likely involved in the transcriptomic reprogramming underlying locule gel formation.

- Furthermore, the manuscript only partially elucidates the specific contribution of *SIMBP3* and its closest homolog *SIAGL11* and their possible interaction in controlling plant and fruit size, and seed development; for example, the discrepancy between the normal vegetative and fruit phenotype of *SIAGL11* silenced plants (Huang et al., 2017) and the reduced vegetative growth and fruit size of the double *SIMBP3-SIAGL11-KO* in the MicroTom background is not discussed. Their respective role in fruit firmness should also be commented upon considering that overexpression of both genes results in an early decrease in firmness.

Answer: This issue certainly needs further clarification. *SIAGL11* and *SIMBP3* are not expressed in the same tissue types as revealed by promoter-GUS expression analyses showing that *SIMBP3* displays a strong expression in locule gel and in funiculus structures, while *SIAGL11* expression is mainly restricted to seeds. These divergent tissue specific expression patterns are sufficient to explain the discrepancy between the phenotypes displayed by the lines silenced in one or another of the two genes. On the other hand, it is not surprising that the overexpression of any of these genes with a constitutive promoter results in similar phenotypes given the high conservation between the two proteins that is indicative of at least partial functional conservation. Indeed, it seems that *SIMBP3* can compensate for *SIAGL11* deficiency for seed formation as the silencing of this latter gene fails to result in any visible phenotype.

-Genome-wide transcriptomic and ChIP analyses that are well conducted and coupled, represent a powerful method to obtain information about the regulatory activity of

SIMBP3 in the process of locular tissue formation. Using this approach, 450 genes were identified that represent putative targets of SIMBP3 and show differential expression in SIMB3-KO locular tissue. However, the authors focused mainly on cell-wall related genes that represent only a minor portion of the whole set. Since the effects on transcription of gene involved in cell wall metabolism has already been observed (Zhang et al., 2019) in SIMBP3 silenced AC plants, I think it would have been interesting to extend an in-depth analysis to other categories of genes as well.

Answer: Based on both RNAseq and ChIP-seq data, the cell wall-related genes are obviously high candidates to play a role in locule gel formation. This motivated our focus on these genes, but we agree with the Reviewer here that among the DEGs those putatively involved in transcriptional regulation and in cell division are also likely to be very important in determining inner tissue structure and fruit texture. It is however, difficult to address into details all gene categories within the framework of this paper. Nevertheless, as suggested by this Reviewer, we performed a functional classification of 450 potential *SIMBP3* direct targets using MapMan software. Overall, 61 genes belonging to 16 different transcription factor families are DEGs and at the same time putative targets of *SIMBP3* as revealed by ChIP-seq (Supplemental Table 2), consistent with the substantial transcriptomic reprogramming observed in the *SIMBP3*-KO lines.

-Overall, I think that although this work provides a lot of information obtained from well-planned and well-conducted experiments, the data presented offer limited novel findings that only partially address the molecular mechanisms underlying the action of SIMBP3 on locular tissue development.

Answer: In addition to the line of arguments elaborated above, we provide further analysis of the cells composing the locular tissue bringing new insight on how *SIMBP3* is influencing the differentiation processes that lead to All-Flesh (*SIMBP3*-KO) or to liquefied gel types (WT). Our data show that over-expanded cells start to appear at early stages (6 DPA) in the emerging locular tissue of WT fruit, concomitant with the appearance of liquefied gel. These large cells become more frequent in number at later stages while continuing to expand leading to disintegrated/fused cells that are not seen in *SIMBP3*-KO fruit. Combined ChIP-seq and RNAseq identified a suite of genes related to cell cycle, cell division and endoreduplication that are putative direct targets of SIMBP3 (Supplementary Dataset 8). These genes may play key role in the differentiation of locular gel tissue and in determining the formation of over-expanded and disintegrated fused cells characteristic of locular gel. As such, these data provide new insight on the mechanisms underlying the action of *SIMBP3* and bring a significant added value to our manuscript, thus adding to the novelty of our findings.

Specific point

-Because the 405- bp deletion in the upstream region of the 5' UTR of SIBMP3 causes a reduction in the transcript level of the gene, it would be interesting to analyze this region for the presence of regulatory element.

Answer: We agree with reviewer's recommendation. We performed *in silico* search (http://plantregmap.gao-lab.org/binding_site_prediction.php) for putative *Cis*-elements within the 405 bp sequence deleted in the promoter of *SIMBP3* in All-flesh cultivars. This analysis identified 8 different types of putative *Cis*-elements, including conserved DNA binding elements for bZIP, C2H2, CPP, GRAS, HD-ZIP, NAC, Nin-Like, WRKY (Supplemental Figure 2d).

-To confirm the finding that the alterations in fruit size and seeds observed in the SIBPM3 silenced AC lines are directly related to the lack of the SIAGL11 gene, it would be appropriate to downregulate SIMBP3 in a different "Gel-Rich" accession defective for AGL11 expression (es MO035).

Answer: We agree with the reviewer that this will bring additional evidence for the detrimental phenotypes induced by dual *SIMBP3/SIAGL11* mutation. However, this will require a tomato genetic transformation that will take more than one year to select stable homozygous lines for the mutation. We have already generated double-KO in MicroTom lines and selected more than 8 independent lines that consistently validate the detrimental phenotypes on plant and fruit growth and on seeds. On the other hand, the commercial All-flesh cultivars used in the present study are all hybrid lines and the knock-out of *SIAGL11* in these lines is likely to give rise to segregating T1 and T2 generations with progenies being different from the parental lines.

-Complementation experiments performed with overexpression of SIMBP3 and SIAGL11 in *Arabidopsis thaliana* do not seem of much interest for the main objective of this work.

Answer: We understand the reviewer's suggestion. We have now removed this figure, although, we think it brings some interesting insight into the extent of functional conservation among class D MADS genes between the model plant *Arabidopsis* bearing siliques and the tomato that produces fleshy fruit.

Reviewer #3:

Huang et al present new information on a gene, MBP3, centrally important in tomato locule formation. They affirm gene function through altered expression and knock out of the MBP3 gene and characterization of a natural promoter mutation that reduces expression. They further demonstrate that MBP3 influences softening through effects that initiate well before ripening begins. Finally, they clarify that seed development and vegetative phenotypes reported in a prior functional characterization of MBP3 were performed in a genotype carrying a natural mutation in the closely related AGL11 gene and result from knock out of both genes. This data is important in furthering our understanding of tomato locule development and provides novel insights into determinants influencing softening. A number of issues should be clarified.

-The introduction briefly describes the formation of locule tissue as parenchymatous cells that have been described as giant cells by others. The cells forming the eventual locule stop dividing early in development and expand considerably. What was not described is how MBP3 is influencing this process. Are the cells in aff or MBP3 KO lines giant cells that fail to liquify or are they parenchyma cells more similar to the pericarp? The microscopy suggests they are more like the pericarp but not exactly the same. A small fruited genotype is used while prior descriptions of locule development may have been describing larger fruit. Does the expansion of the locule cells occur in this genotype? The manuscript proports to describe the role of MBP3 in locule formation but actually provides little insight into the cellular changes that result from altered MBP3 expression. Is the process of cellular proliferation and expansion altered or is the role only in conferring cellular liquefaction? With little effort some images of the larger fruited aff genotypes could help clarify what role if any MBP3 is playing beyond cellular liquefaction.

Answer: We thank the reviewer for raising this important point and we agree that further analysis of the cells composing the locular tissue may provide some significant insight on how *SIMBP3* is influencing the differentiation process that leads either to All-flesh (*SIMBP3-KO*) or to liquefied gel types (WT). Therefore, we performed additional examination of the cell types forming the locular tissue in both *SIMBP3-KO* and WT lines using toluidine blue staining as efficient mean to discriminate between liquefied and non-liquefied locular tissue. The data show that the locular tissue originating from the placenta (as reported by Lemaire-Chamley et al, 2005) starts to show over-expanded cells in WT fruit as early as 6 and 9 DPA, concomitant with the appearance of locular gel (Fig. 7a). At 6 and 9 DPA, WT fruit display significantly lower number of cells per mm² than *SIMBP3-KO* lines (Fig. 7b). Consistently, the average area of individual cells in the locular tissue is dramatically bigger in WT than in *SIMBP3-KO* fruit (Fig. 7c). Moreover, WT fruit exhibit a high number of what they look like disintegrated-fused cells that are not seen in the locular tissue of *SIMBP3-KO* fruit (Fig. 7a). Interestingly, RNAseq

performed on locular tissue at 10-DPA showed that among the genes differentially expressed between WT and *SIMBP3-KO* lines, 114 DEGs are related to cell cycle and cell division (Supplementary Dataset 7 and 8). Notably, among the cell division-related genes that are differentially expressed and being at the same time putative *SIMBP3* targets, based on the ChIP-seq data, several markers of endoreduplication genes are down-regulated in All-flesh lines. Taken together, these data support the hypothesis that *SIMBP3* is driving the formation of liquefied locular tissue through promoting endoreduplication leading to cell expansion and the appearance of disintegrating cells with thin cell walls.

-In the manuscript introduction, last paragraph, it is noted that All-flesh” types are highly sought after by breeders for processing and in fresh processed products. The relevance to practical use is cited in the abstract and discussion as well. Is this actually true for fresh use varieties? At least the fresh varieties I see in the market do not seem to carry this trait nor do I see it in the fresh processed products with which I am familiar. aff genotypes are available and the mutation can be readily bred and is definitely in some processing varieties. Quality attributes and consumer preferences for tomatoes are also influenced by juiciness and the locule was long ago shown to be a reservoir of flavor chemistry in addition to the pulp (pericarp). It would seem aff may be less relevant to fresh types. Clarification in this regard for both processing and fresh types would be useful to clarify this point of practical relevance.

Answer: We certainly understand the reviewer questioning about our statement that the All-flesh trait is also suited for fresh tomato. From what we know after discussing with breeders and tomato producers, this mutation is introduced by seed companies mainly in processing types and not in fresh types. However, the feedback we had from the breeders is that introducing this trait in fresh type tomato might be beneficial to increase the diversity of cultivars in the market and also to reduce sandwiches and hamburgers sogginess. They also did explain that in many cases they were unsuccessful to produce new commercial tomato cultivars for fresh use because of the detrimental effect they obtain when crossing their parental elite lines likely because they bear the *SIAGL11* deletion. Anyway, as suggested by the Reviewer, we removed all references to the All-flesh trait being sought after for fresh use tomato.

-Figure 1 d. Complementation of the aff lines with MBP3 under its endogenous promoter provides partial complementation on all genotypes tested while 35S results in what appears to be a normal liquid locule. Could this indicate that all needed regulatory sequences are not present in the promoter sequences used?

Answer: The reviewer is right, we cannot rule out the possibility that the *SIMBP3* promoter we used for the complementation experiments is incomplete and that some the needed regulatory sequences are missing. Indeed, the *proSIMBP3::SIMBP3* construct

used for complementation is 2.7 kb long but contains 1.5 kb corresponding to an intron embedded within the 5'UTR that is anticipated to be spliced. Also, the complementation is assessed with R0 lines hemizygous for the transgene because the All-flesh commercial cultivars are all hybrid lines that segregate in the next generations. It might be possible that the hemizygous status of the transgene gives insufficient expression levels with *proSIMBP3::SIMBP3*, in contrast to the *35S::SIMBP3* that provides sufficient expression levels.

-It is stated that softening occurs as early as 10 d. Is it more accurate to note that it could be influenced even earlier but 10 d was the first time point examined? Or were there earlier time points with no change tested, and if so such data should be included. This comes up in reference to OE lines and softening at 10 days as well. I imagine looking at earlier stages is perhaps difficult but can anything be said about when the softening differential due to MBP3 can be first identified in normally expressing or OX fruit? If not the text should be modified accordingly.

Answer: As shown in Figure 2c, significant change in firmness is clearly observed at 10-DPA for both KO and OX lines compared to WT. We didn't assess firmness at earlier stages because it is very difficult to handle such a small fruit. We then used Toluidine blue staining as mean to discriminate between the liquefied and non-liquefied locular tissues. The new data presented in Supplementary Figure 9 indicate that in WT fruit the process of locular gel formation is initiated at 8-DPA or maybe earlier (see 6 -DPA in Fig.7a and Supplementary Fig.9) and then becomes more obvious at later stages as shown for 10 and 30 DPA. By contrast, this type of liquefied locular tissue is not present in *SIMBP3-KO* fruit even at 30 DPA.

-Expression of MBP3 is presented using relative terms with aff fruit used as reference. This gives minimal indication of the effect on the deletion on MBP3 expression. Is MBP3 still expressed at any appreciable level in aff fruit as a result of the deletion? In the discussion it is stated that aff lines retain significant expression of MBP3 but one never gets a clear picture of what this significant level is. Based on Fig 2c it appears that is about half of WT but one gets no sense of WT expression levels. Perhaps they could provide insight from TomExpress. Also, is the expression in Fig 2c from locule tissue, whole fruit? The figure does not indicate the tissue used and the legend does not clarify.

Answer: The expression levels of *SIMBP3* in MT and in All-flesh lines were assessed using whole fruit tissues. *SIMBP3* transcript accumulation was calculated based on the transcript levels of the Actin internal reference gene in each line. Then, the outcome of the qPCR data for each line were compared to those of MT to assess change significance using *t*-test. We adopted this approach because there is no reference WT line to use for the All-flesh fruit as they are hybrids and made with parental lines that are unknown for us. Hope this clarifies the issue.

-Line 132 the adjective “subtle” is used describing dosage effect. Is the right terminology? Given the gradation of phenotypes shown in Sup fig 5a perhaps “linear” or “parallel” are more appropriate descriptors?

Answer: We agree with the reviewer and have changed this in the text.

-Lines 176-178. This text is unclear to me. Are they saying that the locule becomes jelly at the 10 DPA stage in WT of the fruit shown in Fig 2 a? The images appear to indicate this is not occurring until 30 DPA. Or is the intent to indicate that the tissue that becomes the liquified gel can be differentiated at this stage? Fig 2 b does indicate lighter staining of WT at 10 DPA but this appears to be much less than at 30 DPA and at least by appearances similar to some of the staining in the supplemental figures where the locules are presented as more solid. In short, the baseline of locule change in the reference genotype should be better described as it is the basis of many of the observations presented.

Answer: We provide now additional data (Supplementary Fig.9 and Fig.7a) to clarify this issue. As mentioned above, we used Toluidine blue staining as efficient mean to discriminate between the liquefied and non-liquefied locular tissues. In the reference WT fruit the process of locular gel formation is initiated at 8-DPA (Supplementary Fig.9), if not earlier, and then becomes more obvious at later stages. By contrast, Toluidine blue staining indicated that the liquefied tissue never appears in the locules of *SIMBP3*-KO fruit.

-Figure 2a shows some interesting phenotypes not mentioned in the manuscript. *AGL11*-KO are clearly liquifying their locules early at 20 DPA where this is not apparent in the WT until 30 DPA. Could *AGL11* be providing a repressive effect? In which case *MBP3* is still the primary regulator as described but *AGL11* may also have a role?

Answer: Careful examination of hundreds of fruits from plants cultivated in a time period exceeding three years, did not allow to seeing clear difference between WT and *SIAGL11*-KO lines with regard to locular gel. The picture the Reviewer is referring to is likely misleading and doesn't reflect the reality. We now changed the picture to better reflect what has been consistently observed. It is worth to mention that it happens that different lines with different levels of the expression of the transgene may present some subtle differences but not really discriminating phenotypes.

-*MBP3* and *AGL11* OX are presented as having similar phenotypes but it appears the *ATL11* OX has seeds in the BR+3 fruit though not in other stages. It appears that the pericarp tissue in 10-30 DPA OX fruit has also become liquified in MG and later stages. Is that true? The dual and OX images are difficult to see. While difficult to describe there appear to be differences between *MBP3* and *AGL11* OX fruit with OX *MBP3* presenting red, light yellow and deep yellow layers (outside to inside) while *AGL11* OX presents

just two layers, deep red and deep yellow. In this regard, images as in b for stages later than 30DPA might be helpful as would larger and clearer images as in Sup Fig 5.

Answer: We fully understand the reviewer comment. The *SIAGL11*-OX fruit never set seeds unless they are manually pollinated; which we do in many cases to maintain the lines. It is possible that the picture presented in the initial Fig. 2a corresponds to one of those fruit obtained by manual pollination. The picture shown in initial Fig. 2a has been changed now to reflect what we always observed. Moreover, we couldn't find any significant difference between the *SIMBP3*-OX and *SIAGL11*-OX fruit which motivated our decision not to go for further characterization of these fruit.

We are providing a picture showing *SIMBP3*-OX and *SIAGL11*-OX fruit at stages later than 30 DPA (MG, BR+3 and BR+8) to hopefully convince the Reviewer that they don't have seeds. The structures visible at the periphery of the placenta correspond to vascular veins but not to seeds. We don't think, however, there is a need to add this picture as supplemental Figure.

With regard to the ectopic modification of the pericarp in OX lines, the Reviewer is may be right, the pericarp tissue in both *SIAGL11*-OX and *SIMBP3*-OX fruit tends to have an appearance that resembles the liquefied gel. However, it is difficult to draw solid conclusion with OX of a transgene driven by 35S constitutive promoter that is anticipated to express the protein in tissues and at developmental stages where it is not expressed normally.

-Why is it stated in lines 189-191 that the two D class genes may have similar function when the KO lines point out they do not? It is stated explicitly in the discussion that they have different functions. This text should be clarified

Answer: It is true that the knockout of the two genes leads to completely divergent phenotypes which suggest divergent functionalities. Nevertheless, the phenotypic differences between the *SIMBP3*-KO and *SIAGL11*-KO lines do not necessarily mean

that the two proteins are functionally divergent. The ability of both genes to compensate the All-flesh phenotype and the similarity of the phenotypes of their OX lines suggest that are capable to have similar functions. The two genes are expressed in different territories except in seeds where GUS reporter driven by their promoter shows convergent expression in seed structure. It cannot be ruled out that *SIMBP3* may compensate for *SIAGL11* mutation which explain the absence of visible phenotypes in *SIAGL11*-KO lines. Altogether, these data support the notion that the two class D tomato proteins may share some functionalities.

Reviewers' Comments:

Reviewer #1:

Remarks to the Author:

The authors revised manuscript satisfactorily and I learned that the constitutive expression of SIAGL11 in a SIMBP3-KO mutant restored locular development, which strongly supports the authors findings, the functional redundancy of two homologues and the cause of the difference between an AC-RNAi fruit and a KO-fruit. I satisfied the revised manuscript and have no more comments or suggestions.

Reviewer #2:

Remarks to the Author:

The authors have convincingly responded to my criticisms and suggestions. By adding new experiments showing the histological difference in fruit development between WT and SIMB3-KO, they have supported the novel finding that the process determining fruit firmness is regulated at a very early stage of fruit development. They also provide new insights into unraveling the role of SIMB3 in the transcriptional reprogramming underlying the formation of locular gel tissue.

Reviewer #3:

Remarks to the Author:

I am grateful for the authors response to my prior review. The inclusion of the new data describing the effects of MBP3 on the progression of locule development helped clarify many prior concerns and the overall presentation. I have no further comments.

REVIEWER COMMENTS

Reviewer #1:

This manuscript describes identification of factors defining locular tissue development in tomato fruit. SIMBP3 have been already indicated to be involved in the locular tissue development by an RNAi experiment, but the transformants from the cultivar Ailsa Craig showed several side effects other than defect of locular tissue development, which was not found in commercial “All-flesh” cultivars. The results might have confused the argument of the regulation for locular tissue development. In this study, the authors provide unambiguous evidence that SIMBP3 determines locular tissue development and alleles of SIAGL11, a paralog of SIMBP3, affect the side effects found in the RNAi tomato from Ailsa Craig. The authors found a mutation in SIMBP3 among All-flesh cultivars and a long deletion in the SIAGL11 locus in Ailsa Craig. By CRISPR/Cas9 mediated knockout and RNAi mediated knockdown experiments of either or both SIMBP3 and SIAGL11 indicates partial redundant activities of the two transcription factors but the experiments provided clear-cut evidences that locular tissue development is exclusively dependent on SIMBP3. Over-expression analyses, and other physiological and transcriptome or ChIP analyses strongly support the finding. This study examined almost all possibilities and the results likely support the conclusion perfectly, so I have no doubt or no comment to the conclusion of this manuscript. In this reviewing, the argument for this study may be whether the study provides novelties enough for the standard of this journal. As described by the authors, SIMBP3 has been known to be involved in locular tissue development, although the previous study, the RNAi experiment for Ailsa Craig, did not account for the mechanisms of the all-flesh phenotype found in commercial cultivars. Therefore, it might be easy to focus SIMBP3 as a start point of the study revealing the mechanism for the locular tissue development. However, the experimental designs to find another “cryptic” mutation in Ailsa Craig and determine SIMBP3 as the only factor for the phenotype in the WT background are elegant, and to achieve the simple conclusion, many other possibilities were excluded by developing various kinds of knockout mutants or knockdown/over-expression transformants. In addition, the finding will contribute practical breeding programs for processing tomato cultivars. Therefore, I believe this manuscript has enough originality and novelty. This is just a comment for the authors, if SIAGL11 with a constitutive promoter is overexpressed in all-flesh cultivars or a SIMBP3-KO line, is locular tissue development restored by complementing the SIMBP3 deficiency? The possibility is described in line 4 of page 9 but I could not find the result in this manuscript.

Reviewer #2:

The manuscript by Huang et al., entitled “Cryptic variation of two MADS-box genes resolves the causal factor of phenotypic variation of locular tissue in tomato fruit thus defining optimal breeding strategies” provides a clear association between the “All flesh” phenotype in tomato and a 405bp deletion in the promoter region of a class-D MADS-box gene, SIMBP3, causing downregulation of the gene. Analysis of the fruit phenotype shown by the SIMBP3-RNAi lines with different degree of silencing also demonstrates that the expression level of the gene correlates with the severity of the “All flesh” trait. The “All-flesh” trait obtained by SIMBP3 knock-out/silencing confers to the fruits enhanced firmness and longer shelf life.

The manuscript also provides evidence that overexpressing SIMBP3 can convert “All Flesh” fruits to jelly type fruits. In addition, it associates reduction in fruit size and seed malformation observed when SIMBP3 is silenced in the cultivar Ailsa Craig with the knock-out of its closest homolog SIAGL11, due to a deletion in the gene sequence. Application of RNA-seq analysis to locular tissue and fruits of SIMBP3 -KO plants coupled with ChIP-seq, allows the identification of differentially expressed genes that are putatively regulated by SIMBP3 at the transcriptional level. These analyses reveal that genes coding for enzymes involved in cell wall modification are key targets of SIMBP3.

- This work provides a robust demonstration that SIMBP3 is a master regulator of locular tissue development in tomato and thus a powerful tool for tomato breeding. However, these results are only partially novel. Clear indications of the role of SIMBP3 in locular gel tissue formation were presented in the work of Zhang et al, (J Exp Bot 2019) along with the observation of the lack of SIAGL11 expression in the cultivar Ailsa Craig. Several pieces of information regarding the expression pattern of SIMBP3 and SIAGL11, the phenotypic traits of SIAGL11-RNAi e SIAGL11-OE plants, and the impact of SIMBP3 and SIAGL11 on the regulation of cell wall- related genes seems to largely confirm previous observations (Zhang et al., 2019, Huang et al. J Exp Bot 2017).

- Furthermore, the manuscript only partially elucidates the specific contribution of SIMBP3 and its closest homolog SIAGL11 and their possible interaction in controlling plant and fruit size, and seed development; for example, the discrepancy between the normal vegetative and fruit phenotype of SIAGL11 silenced plants (Huang et al., 2017) and the reduced vegetative growth and fruit size of the double SIMBP3-SIAGL11-KO in the MicroTom background is not discussed. Their respective role in fruit firmness should also be commented upon considering that overexpression of both genes results in an early decrease in firmness.

-Genome-wide transcriptomic and ChIP analyses that are well conducted and coupled, represent a powerful method to obtain information about the regulatory activity of SIMBP3 in the process of locular tissue formation. Using this approach, 450 genes were identified that represent putative targets of SIMBP3 and show differential expression in

SIMB3-KO locular tissue. However, the authors focused mainly on cell-wall related genes that represent only a minor portion of the whole set. Since the effects on transcription of gene involved in cell wall metabolism has already been observed (Zhang et al., 2019) in SIMBP3 silenced AC plants, I think it would have been interesting to extend an in-depth analysis to other categories of genes as well.

-Overall, I think that although this work provides a lot of information obtained from well-planned and well-conducted experiments, the data presented offer limited novel findings that only partially address the molecular mechanisms underlying the action of SIMBP3 on locular tissue development.

Specific point

-Because the 405- bp deletion in the upstream region of the 5' UTR of SIBMP3 causes a reduction in the transcript level of the gene, it would be interesting to analyze this region for the presence of regulatory element.

-To confirm the finding that the alterations in fruit size and seeds observed in the SIBPM3 silenced AC lines are directly related to the lack of the SIAGL11 gene, it would be appropriate to downregulate SIMBP3 in a different “Gel-Rich” accession defective for AGL11 expression (es MO035).

-Complementation experiments performed with overexpression of SIMBP3 and SIAGL11 in *Arabidopsis thaliana* do not seem of much interest for the main objective of this work.

Reviewer #3:

Huang et al present new information on a gene, MBP3, centrally important in tomato locule formation. They affirm gene function through altered expression and knock out of the MBP3 gene and characterization of a natural promoter mutation that reduces expression. They further demonstrate that MBP3 influences softening through effects that initiate well before ripening begins. Finally, they clarify that seed development and vegetative phenotypes reported in a prior functional characterization of MBP3 were performed in a genotype carrying a natural mutation in the closely related AGL11 gene and result from knock out of both genes. This data is important in furthering our understanding of tomato locule development and provides novel insights into determinants influencing softening. A number of issues should be clarified.

-The introduction briefly describes the formation of locule tissue as parenchymatous cells that have been described as giant cells by others. The cells forming the eventual locule

stop dividing early in development and expand considerably. What was not described is how MBP3 is influencing this process. Are the cells in aff or MBP3 KO lines giant cells that fail to liquify or are they parenchyma cells more similar to the pericarp? The microscopy suggests they are more like the pericarp but not exactly the same. A small fruited genotype is used while prior descriptions of locule development may have been describing larger fruit. Does the expansion of the locule cells occur in this genotype? The manuscript purports to describe the role of MBP3 in locule formation but actually provides little insight into the cellular changes that result from altered MBP3 expression. Is the process of cellular proliferation and expansion altered or is the role only in conferring cellular liquefaction? With little effort some images of the larger fruited aff genotypes could help clarify what role if any MBP3 is playing beyond cellular liquefaction.

-In the manuscript introduction, last paragraph, it is noted that All-flesh” types are highly sought after by breeders for processing and in fresh processed products. The relevance to practical use is cited in the abstract and discussion as well. Is this actually true for fresh use varieties? At least the fresh varieties I see in the market do not seem to carry this trait nor do I see it in the fresh processed products with which I am familiar. aff genotypes are available and the mutation can be readily bred and is definitely in some processing varieties. Quality attributes and consumer preferences for tomatoes are also influenced by juiciness and the locule was long ago shown to be a reservoir of flavor chemistry in addition to the pulp (pericarp). It would seem aff may be less relevant to fresh types. Clarification in this regard for both processing and fresh types would be useful to clarify this point of practical relevance.

-Figure 1 d. Complementation of the aff lines with MBP3 under its endogenous promoter provides partial complementation on all genotypes tested while 35S results in what appears to be a normal liquid locule. Could this indicate that all needed regulatory sequences are not present in the promoter sequences used?

-It is stated that softening occurs as early as 10 d. Is it more accurate to note that it could be influenced even earlier but 10 d was the first time point examined? Or were there earlier time points with no change tested, and if so such data should be included. This comes up in reference to OE lines and softening at 10 days as well. I imagine looking at earlier stages is perhaps difficult but can anything be said about when the softening differential due to MBP3 can be first identified in normally expressing or OX fruit? If not the text should be modified accordingly.

-Expression of MBP3 is presented using relative terms with aff fruit used as reference. This gives minimal indication of the effect on the deletion on MBP3 expression. Is MBP3

still expressed at any appreciable level in aff fruit as a result of the deletion? In the discussion it is stated that aff lines retain significant expression of MBP3 but one never gets a clear picture of what this significant level is. Based on Fig 2c it appears that is about half of WT but one gets no sense of WT expression levels. Perhaps they could provide insight from TomExpress. Also, is the expression in Fig 2c from locule tissue, whole fruit? The figure does not indicate the tissue used and the legend does not clarify.

-Line 132 the adjective “subtle” is used describing dosage effect. Is the right terminology? Given the gradation of phenotypes shown in Sup fig 5a perhaps “linear” or “parallel” are more appropriate descriptors?

-Lines 176-178. This text is unclear to me. Are they saying that the locule becomes jelly at the 10 DPA stage in WT of the fruit shown in Fig 2 a? The images appear to indicate this is not occurring until 30 DPA. Or is the intent to indicate that the tissue that becomes the liquified gel can be differentiated at this stage? Fig 2 b does indicate lighter staining of WT at 10 DPA but this appears to be much less than at 30 DPA and at least by appearances similar to some of the staining in the supplemental figures where the locules are presented as more solid. In short, the baseline of locule change in the reference genotype should be better described as it is the basis of many of the observations presented.

-Figure 2a shows some interesting phenotypes not mentioned in the manuscript. AGL11-KO are clearly liquifying their locules early at 20 DPA where this is not apparent in the WT until 30 DPA. Could AGL11 be providing a repressive effect? In which case MBP3 is still the primary regulator as described but AGL11 may also have a role?

-MBP3 and AGL11 OX are presented as having similar phenotypes but it appears the ATL11 OX has seeds in the BR+3 fruit though not in other stages. It appears that the pericarp tissue in 10-30 DPA OX fruit has also become liquified in MG and later stages. Is that true? The dual and OX images are difficult to see. While difficult to describe there appear to be differences between MBP3 and AGL11 OX fruit with OX MBP3 presenting red, light yellow and deep yellow layers (outside to inside) while AGL11 OX presents just two layers, deep red and deep yellow. In this regard, images as in b for stages later than 30DPA might be helpful as would larger and clearer images as in Sup Fig 5.

-Why is it stated in lines 189-191 that the two D class genes may have similar function when the KO lines point out they do not? It is stated explicitly in the discussion that they have different functions. This text should be clarified

RESPONSE TO REVIEWERS

The following are our point-by-point answers to the reviewers' questions and comments.

Reviewer #1:

This manuscript describes identification of factors defining locular tissue development in tomato fruit. SIMBP3 have been already indicated to be involved in the locular tissue development by an RNAi experiment, but the transformants from the cultivar Ailsa Craig showed several side effects other than defect of locular tissue development, which was not found in commercial “All-flesh” cultivars. The results might have confused the argument of the regulation for locular tissue development. In this study, the authors provide unambiguous evidence that SIMBP3 determines locular tissue development and alleles of SIAGL11, a paralog of SIMBP3, affect the side effects found in the RNAi tomato from Ailsa Craig. The authors found a mutation in SIMBP3 among All-flesh cultivars and a long deletion in the SIAGL11 locus in Ailsa Craig. By CRISPR/Cas9 mediated knockout and RNAi mediated knockdown experiments of either or both SIMBP3 and SIAGL11 indicates partial redundant activities of the two transcription factors but the experiments provided clear-cut evidences that locular tissue development is exclusively dependent on SIMBP3. Over-expression analyses, and other physiological and transcriptome or ChIP analyses strongly support the finding. This study examined almost all possibilities and the results likely support the conclusion perfectly, so I have no doubt or no comment to the conclusion of this manuscript. In this reviewing, the argument for this study may be whether the study provides novelties enough for the standard of this journal. As described by the authors, SIMBP3 has been known to be involved in locular tissue development, although the previous study, the RNAi experiment for Ailsa Craig, did not account for the mechanisms of the all-flesh phenotype found in commercial cultivars. Therefore, it might be easy to focus SIMBP3 as a start point of the study revealing the mechanism for the locular tissue development. However, the experimental designs to find another “cryptic” mutation in Ailsa Craig and determine SIMBP3 as the only factor for the phenotype in the WT background are elegant, and to achieve the simple conclusion, many other possibilities were excluded by developing various kinds of knockout mutants or knockdown/over-expression transformants. In addition, the finding will contribute practical breeding programs for processing tomato cultivars. Therefore, I believe this manuscript has enough originality and novelty. This is just a comment for the authors, if SIAGL11 with a constitutive promoter is overexpressed in all-flesh cultivars or a SIMBP3-KO line, is locular tissue development restored by complementing the SIMBP3 deficiency? The possibility is described in line 4 of page 9 but I could not find the result in this manuscript.

Answer: The reviewer is right. We have now added this data as Supplementary Fig 6c, showing that the expression of *SIAGL11* driven by a constitutive promoter in the *SIMBP3*-KO lines restores wild type-like locular gel phenotype.

Reviewer #2:

The manuscript by Huang et al., entitled “Cryptic variation of two MADS-box genes resolves the causal factor of phenotypic variation of locular tissue in tomato fruit thus defining optimal breeding strategies” provides a clear association between the “All flesh” phenotype in tomato and a 405bp deletion in the promoter region of a class-D MADS-box gene, *SIMBP3*, causing downregulation of the gene. Analysis of the fruit phenotype shown by the *SIMBP3*-RNAi lines with different degree of silencing also demonstrates that the expression level of the gene correlates with the severity of the “All flesh” trait. The “All-flesh” trait obtained by *SIMBP3* knock-out/silencing confers to the fruits enhanced firmness and longer shelf life.

The manuscript also provides evidence that overexpressing *SIMBP3* can convert “All Flesh” fruits to jelly type fruits. In addition, it associates reduction in fruit size and seed malformation observed when *SIMBP3* is silenced in the cultivar Ailsa Craig with the knock-out of its closest homolog *SIAGL11*, due to a deletion in the gene sequence. Application of RNA-seq analysis to locular tissue and fruits of *SIMBP3* -KO plants coupled with CHIP-seq, allows the identification of differentially expressed genes that are putatively regulated by *SIMBP3* at the transcriptional level. These analyses reveal that genes coding for enzymes involved in cell wall modification are key targets of *SIMBP3*.

- This work provides a robust demonstration that *SIMBP3* is a master regulator of locular tissue development in tomato and thus a powerful tool for tomato breeding. However, these results are only partially novel. Clear indications of the role of *SIMBP3* in locular gel tissue formation were presented in the work of Zhang et al, (J Exp Bot 2019) along with the observation of the lack of *SIAGL11* expression in the cultivar Ailsa Craig. Several pieces of information regarding the expression pattern of *SIMBP3* and *SIAGL11*, the phenotypic traits of *SIAGL11*-RNAi e *SIAGL11*-OE plants, and the impact of *SIMBP3* and *SIAGL11* on the regulation of cell wall- related genes seems to largely confirm previous observations (Zhang et al., 2019, Huang et al. J Exp Bot 2017).

Answer: We respectfully disagree with the reviewer here. One major novelty of our finding is to uncover that the severe detrimental phenotypes on plant growth, fruit size and seed formation are due to dual mutation of both *SIMBP3* and *SIAGL11* which has never been reported before. These vegetative and reproductive growth phenotypes are not only due to *SIMBP3* as reported in Zhang et al. 2019. Indeed, the complete knock-out of *SIMBP3* results in detrimental effects on vegetative and reproductive growth only in a genetic background defective in *SIAGL11*. Our data reveal for the first time that several

tomato genotypes like Ailsa Craig have a big deletion (12 kb) at the *SIAGL11* locus. This is the causal factor for the phenotypic divergence between the phenotypes we describe for *SIMBP3*-silenced lines and those reported by Zhang et al. 2019. This natural mutation at the *SIAGL11* locus underlies the cryptic genetic variation impacting the phenotypes of *SIMBP3* mutation in different genetic backgrounds. We think this information is instrumental for designing efficient breeding strategies aiming to gain the All-flesh trait.

A second major novel insight brought by our study is related to the process determining fruit firmness and texture. So far, the deciphering of the components underlying the softening process has been mostly addressed by focusing on late stages of fruit development, namely the ripening phase, however, our data support the notion that a large component of texture and firmness of ripe fruit is determined at early pre-ripening stages, concomitant with the initiation of inner tissues differentiation.

Another novel insight provided by our data is the identification of a number of genes that are direct targets of *SIMBP3*, including cell-wall, cell division and endoreduplication-related genes as well as a number of transcriptional regulators that are likely involved in the transcriptomic reprogramming underlying locule gel formation.

- Furthermore, the manuscript only partially elucidates the specific contribution of *SIMBP3* and its closest homolog *SIAGL11* and their possible interaction in controlling plant and fruit size, and seed development; for example, the discrepancy between the normal vegetative and fruit phenotype of *SIAGL11* silenced plants (Huang et al., 2017) and the reduced vegetative growth and fruit size of the double *SIMBP3-SIAGL11-KO* in the MicroTom background is not discussed. Their respective role in fruit firmness should also be commented upon considering that overexpression of both genes results in an early decrease in firmness.

Answer: This issue certainly needs further clarification. *SIAGL11* and *SIMBP3* are not expressed in the same tissue types as revealed by promoter-GUS expression analyses showing that *SIMBP3* displays a strong expression in locule gel and in funiculus structures, while *SIAGL11* expression is mainly restricted to seeds. These divergent tissue specific expression patterns are sufficient to explain the discrepancy between the phenotypes displayed by the lines silenced in one or another of the two genes. On the other hand, it is not surprising that the overexpression of any of these genes with a constitutive promoter results in similar phenotypes given the high conservation between the two proteins that is indicative of at least partial functional conservation. Indeed, it seems that *SIMBP3* can compensate for *SIAGL11* deficiency for seed formation as the silencing of this latter gene fails to result in any visible phenotype.

-Genome-wide transcriptomic and ChIP analyses that are well conducted and coupled, represent a powerful method to obtain information about the regulatory activity of *SIMBP3* in the process of locular tissue formation. Using this approach, 450 genes were identified that represent putative targets of *SIMBP3* and show differential expression in

SIMBP3-KO locular tissue. However, the authors focused mainly on cell-wall related genes that represent only a minor portion of the whole set. Since the effects on transcription of gene involved in cell wall metabolism has already been observed (Zhang et al., 2019) in SIMBP3 silenced AC plants, I think it would have been interesting to extend an in-depth analysis to other categories of genes as well.

Answer: Based on both RNAseq and ChIP-seq data, the cell wall-related genes are obviously high candidates to play a role in locule gel formation. This motivated our focus on these genes, but we agree with the Reviewer here that among the DEGs those putatively involved in transcriptional regulation and in cell division are also likely to be very important in determining inner tissue structure and fruit texture. It is however, difficult to address into details all gene categories within the framework of this paper. Nevertheless, as suggested by this Reviewer, we performed a functional classification of 450 potential *SIMBP3* direct targets using MapMan software. Overall, 61 genes belonging to 16 different transcription factor families are DEGs and at the same time putative targets of *SIMBP3* as revealed by ChIP-seq (Supplemental Table 2), consistent with the substantial transcriptomic reprogramming observed in the *SIMBP3*-KO lines.

-Overall, I think that although this work provides a lot of information obtained from well-planned and well-conducted experiments, the data presented offer limited novel findings that only partially address the molecular mechanisms underlying the action of *SIMBP3* on locular tissue development.

Answer: In addition to the line of arguments elaborated above, we provide further analysis of the cells composing the locular tissue bringing new insight on how *SIMBP3* is influencing the differentiation processes that lead to All-Flesh (*SIMBP3*-KO) or to liquefied gel types (WT). Our data show that over-expanded cells start to appear at early stages (6 DPA) in the emerging locular tissue of WT fruit, concomitant with the appearance of liquefied gel. These large cells become more frequent in number at later stages while continuing to expand leading to disintegrated/fused cells that are not seen in *SIMBP3*-KO fruit. Combined ChIP-seq and RNAseq identified a suite of genes related to cell cycle, cell division and endoreduplication that are putative direct targets of *SIMBP3* (Supplementary Dataset 8). These genes may play key role in the differentiation of locular gel tissue and in determining the formation of over-expanded and disintegrated fused cells characteristic of locular gel. As such, these data provide new insight on the mechanisms underlying the action of *SIMBP3* and bring a significant added value to our manuscript, thus adding to the novelty of our findings.

Specific point

-Because the 405- bp deletion in the upstream region of the 5' UTR of SIBMP3 causes a reduction in the transcript level of the gene, it would be interesting to analyze this region for the presence of regulatory element.

Answer: We agree with reviewer's recommendation. We performed *in silico* search (http://plantregmap.gao-lab.org/binding_site_prediction.php) for putative *Cis*-elements within the 405 bp sequence deleted in the promoter of *SIMBP3* in All-flesh cultivars. This analysis identified 8 different types of putative *Cis*-elements, including conserved DNA binding elements for bZIP, C2H2, CPP, GRAS, HD-ZIP, NAC, Nin-Like, WRKY (Supplemental Figure 2d).

-To confirm the finding that the alterations in fruit size and seeds observed in the SIBPM3 silenced AC lines are directly related to the lack of the SIAGL11 gene, it would be appropriate to downregulate SIMBP3 in a different "Gel-Rich" accession defective for AGL11 expression (es MO035).

Answer: We agree with the reviewer that this will bring additional evidence for the detrimental phenotypes induced by dual *SIMBP3/SIAGL11* mutation. However, this will require a tomato genetic transformation that will take more than one year to select stable homozygous lines for the mutation. We have already generated double-KO in MicroTom lines and selected more than 8 independent lines that consistently validate the detrimental phenotypes on plant and fruit growth and on seeds. On the other hand, the commercial All-flesh cultivars used in the present study are all hybrid lines and the knock-out of *SIAGL11* in these lines is likely to give rise to segregating T1 and T2 generations with progenies being different from the parental lines.

-Complementation experiments performed with overexpression of SIMBP3 and SIAGL11 in *Arabidopsis thaliana* do not seem of much interest for the main objective of this work.

Answer: We understand the reviewer's suggestion. We have now removed this figure, although, we think it brings some interesting insight into the extent of functional conservation among class D MADS genes between the model plant *Arabidopsis* bearing siliques and the tomato that produces fleshy fruit.

Reviewer #3:

Huang et al present new information on a gene, MBP3, centrally important in tomato locule formation. They affirm gene function through altered expression and knock out of the MBP3 gene and characterization of a natural promoter mutation that reduces expression. They further demonstrate that MBP3 influences softening through effects that initiate well before ripening begins. Finally, they clarify that seed development and vegetative phenotypes reported in a prior functional characterization of MBP3 were performed in a genotype carrying a natural mutation in the closely related AGL11 gene and result from knock out of both genes. This data is important in furthering our understanding of tomato locule development and provides novel insights into determinants influencing softening. A number of issues should be clarified.

-The introduction briefly describes the formation of locule tissue as parenchymatous cells that have been described as giant cells by others. The cells forming the eventual locule stop dividing early in development and expand considerably. What was not described is how MBP3 is influencing this process. Are the cells in aff or MBP3 KO lines giant cells that fail to liquify or are they parenchyma cells more similar to the pericarp? The microscopy suggests they are more like the pericarp but not exactly the same. A small fruited genotype is used while prior descriptions of locule development may have been describing larger fruit. Does the expansion of the locule cells occur in this genotype? The manuscript proports to describe the role of MBP3 in locule formation but actually provides little insight into the cellular changes that result from altered MBP3 expression. Is the process of cellular proliferation and expansion altered or is the role only in conferring cellular liquefaction? With little effort some images of the larger fruited aff genotypes could help clarify what role if any MBP3 is playing beyond cellular liquefaction.

Answer: We thank the reviewer for raising this important point and we agree that further analysis of the cells composing the locular tissue may provide some significant insight on how *SIMBP3* is influencing the differentiation process that leads either to All-flesh (*SIMBP3-KO*) or to liquefied gel types (WT). Therefore, we performed additional examination of the cell types forming the locular tissue in both *SIMBP3-KO* and WT lines using toluidine blue staining as efficient mean to discriminate between liquefied and non-liquefied locular tissue. The data show that the locular tissue originating from the placenta (as reported by Lemaire-Chamley et al, 2005) starts to show over-expanded cells in WT fruit as early as 6 and 9 DPA, concomitant with the appearance of locular gel (Fig. 7a). At 6 and 9 DPA, WT fruit display significantly lower number of cells per mm² than *SIMBP3-KO* lines (Fig. 7b). Consistently, the average area of individual cells in the locular tissue is dramatically bigger in WT than in *SIMBP3-KO* fruit (Fig. 7c). Moreover, WT fruit exhibit a high number of what they look like disintegrated-fused cells that are not seen in the locular tissue of *SIMBP3-KO* fruit (Fig. 7a). Interestingly, RNAseq

performed on locular tissue at 10-DPA showed that among the genes differentially expressed between WT and *SIMBP3-KO* lines, 114 DEGs are related to cell cycle and cell division (Supplementary Dataset 7 and 8). Notably, among the cell division-related genes that are differentially expressed and being at the same time putative *SIMBP3* targets, based on the ChIP-seq data, several markers of endoreduplication genes are down-regulated in All-flesh lines. Taken together, these data support the hypothesis that *SIMBP3* is driving the formation of liquefied locular tissue through promoting endoreduplication leading to cell expansion and the appearance of disintegrating cells with thin cell walls.

-In the manuscript introduction, last paragraph, it is noted that All-flesh” types are highly sought after by breeders for processing and in fresh processed products. The relevance to practical use is cited in the abstract and discussion as well. Is this actually true for fresh use varieties? At least the fresh varieties I see in the market do not seem to carry this trait nor do I see it in the fresh processed products with which I am familiar. aff genotypes are available and the mutation can be readily bred and is definitely in some processing varieties. Quality attributes and consumer preferences for tomatoes are also influenced by juiciness and the locule was long ago shown to be a reservoir of flavor chemistry in addition to the pulp (pericarp). It would seem aff may be less relevant to fresh types. Clarification in this regard for both processing and fresh types would be useful to clarify this point of practical relevance.

Answer: We certainly understand the reviewer questioning about our statement that the All-flesh trait is also suited for fresh tomato. From what we know after discussing with breeders and tomato producers, this mutation is introduced by seed companies mainly in processing types and not in fresh types. However, the feedback we had from the breeders is that introducing this trait in fresh type tomato might be beneficial to increase the diversity of cultivars in the market and also to reduce sandwiches and hamburgers sogginess. They also did explain that in many cases they were unsuccessful to produce new commercial tomato cultivars for fresh use because of the detrimental effect they obtain when crossing their parental elite lines likely because they bear the *SIAGL11* deletion. Anyway, as suggested by the Reviewer, we removed all references to the All-flesh trait being sought after for fresh use tomato.

-Figure 1 d. Complementation of the aff lines with MBP3 under its endogenous promoter provides partial complementation on all genotypes tested while 35S results in what appears to be a normal liquid locule. Could this indicate that all needed regulatory sequences are not present in the promoter sequences used?

Answer: The reviewer is right, we cannot rule out the possibility that the *SIMBP3* promoter we used for the complementation experiments is incomplete and that some the needed regulatory sequences are missing. Indeed, the *proSIMBP3::SIMBP3* construct

used for complementation is 2.7 kb long but contains 1.5 kb corresponding to an intron embedded within the 5'UTR that is anticipated to be spliced. Also, the complementation is assessed with R0 lines hemizygous for the transgene because the All-flesh commercial cultivars are all hybrid lines that segregate in the next generations. It might be possible that the hemizygous status of the transgene gives insufficient expression levels with *proSIMBP3::SIMBP3*, in contrast to the *35S::SIMBP3* that provides sufficient expression levels.

-It is stated that softening occurs as early as 10 d. Is it more accurate to note that it could be influenced even earlier but 10 d was the first time point examined? Or were there earlier time points with no change tested, and if so such data should be included. This comes up in reference to OE lines and softening at 10 days as well. I imagine looking at earlier stages is perhaps difficult but can anything be said about when the softening differential due to MBP3 can be first identified in normally expressing or OX fruit? If not the text should be modified accordingly.

Answer: As shown in Figure 2c, significant change in firmness is clearly observed at 10-DPA for both KO and OX lines compared to WT. We didn't assess firmness at earlier stages because it is very difficult to handle such a small fruit. We then used Toluidine blue staining as mean to discriminate between the liquefied and non-liquefied locular tissues. The new data presented in Supplementary Figure 9 indicate that in WT fruit the process of locular gel formation is initiated at 8-DPA or maybe earlier (see 6 -DPA in Fig.7a and Supplementary Fig.9) and then becomes more obvious at later stages as shown for 10 and 30 DPA. By contrast, this type of liquefied locular tissue is not present in *SIMBP3-KO* fruit even at 30 DPA.

-Expression of MBP3 is presented using relative terms with aff fruit used as reference. This gives minimal indication of the effect on the deletion on MBP3 expression. Is MBP3 still expressed at any appreciable level in aff fruit as a result of the deletion? In the discussion it is stated that aff lines retain significant expression of MBP3 but one never gets a clear picture of what this significant level is. Based on Fig 2c it appears that is about half of WT but one gets no sense of WT expression levels. Perhaps they could provide insight from TomExpress. Also, is the expression in Fig 2c from locule tissue, whole fruit? The figure does not indicate the tissue used and the legend does not clarify.

Answer: The expression levels of *SIMBP3* in MT and in All-flesh lines were assessed using whole fruit tissues. *SIMBP3* transcript accumulation was calculated based on the transcript levels of the Actin internal reference gene in each line. Then, the outcome of the qPCR data for each line were compared to those of MT to assess change significance using *t*-test. We adopted this approach because there is no reference WT line to use for the All-flesh fruit as they are hybrids and made with parental lines that are unknown for us. Hope this clarifies the issue.

-Line 132 the adjective “subtle” is used describing dosage effect. Is the right terminology? Given the gradation of phenotypes shown in Sup fig 5a perhaps “linear” or “parallel” are more appropriate descriptors?

Answer: We agree with the reviewer and have changed this in the text.

-Lines 176-178. This text is unclear to me. Are they saying that the locule becomes jelly at the 10 DPA stage in WT of the fruit shown in Fig 2 a? The images appear to indicate this is not occurring until 30 DPA. Or is the intent to indicate that the tissue that becomes the liquified gel can be differentiated at this stage? Fig 2 b does indicate lighter staining of WT at 10 DPA but this appears to be much less than at 30 DPA and at least by appearances similar to some of the staining in the supplemental figures where the locules are presented as more solid. In short, the baseline of locule change in the reference genotype should be better described as it is the basis of many of the observations presented.

Answer: We provide now additional data (Supplementary Fig.9 and Fig.7a) to clarify this issue. As mentioned above, we used Toluidine blue staining as efficient mean to discriminate between the liquefied and non-liquefied locular tissues. In the reference WT fruit the process of locular gel formation is initiated at 8-DPA (Supplementary Fig.9), if not earlier, and then becomes more obvious at later stages. By contrast, Toluidine blue staining indicated that the liquefied tissue never appears in the locules of *SIMBP3*-KO fruit.

-Figure 2a shows some interesting phenotypes not mentioned in the manuscript. *AGL11*-KO are clearly liquifying their locules early at 20 DPA where this is not apparent in the WT until 30 DPA. Could *AGL11* be providing a repressive effect? In which case *MBP3* is still the primary regulator as described but *AGL11* may also have a role?

Answer: Careful examination of hundreds of fruits from plants cultivated in a time period exceeding three years, did not allow to seeing clear difference between WT and *SIAGL11*-KO lines with regard to locular gel. The picture the Reviewer is referring to is likely misleading and doesn't reflect the reality. We now changed the picture to better reflect what has been consistently observed. It is worth to mention that it happens that different lines with different levels of the expression of the transgene may present some subtle differences but not really discriminating phenotypes.

-*MBP3* and *AGL11* OX are presented as having similar phenotypes but it appears the *ATL11* OX has seeds in the BR+3 fruit though not in other stages. It appears that the pericarp tissue in 10-30 DPA OX fruit has also become liquified in MG and later stages. Is that true? The dual and OX images are difficult to see. While difficult to describe there appear to be differences between *MBP3* and *AGL11* OX fruit with OX *MBP3* presenting red, light yellow and deep yellow layers (outside to inside) while *AGL11* OX presents

just two layers, deep red and deep yellow. In this regard, images as in b for stages later than 30DPA might be helpful as would larger and clearer images as in Sup Fig 5.

Answer: We fully understand the reviewer comment. The *SIAGL11*-OX fruit never set seeds unless they are manually pollinated; which we do in many cases to maintain the lines. It is possible that the picture presented in the initial Fig. 2a corresponds to one of those fruit obtained by manual pollination. The picture shown in initial Fig. 2a has been changed now to reflect what we always observed. Moreover, we couldn't find any significant difference between the *SIMBP3*-OX and *SIAGL11*-OX fruit which motivated our decision not to go for further characterization of these fruit.

We are providing a picture showing *SIMBP3*-OX and *SIAGL11*-OX fruit at stages later than 30 DPA (MG, BR+3 and BR+8) to hopefully convince the Reviewer that they don't have seeds. The structures visible at the periphery of the placenta correspond to vascular veins but not to seeds. We don't think, however, there is a need to add this picture as supplemental Figure.

With regard to the ectopic modification of the pericarp in OX lines, the Reviewer is may be right, the pericarp tissue in both *SIAGL11*-OX and *SIMBP3*-OX fruit tends to have an appearance that resembles the liquefied gel. However, it is difficult to draw solid conclusion with OX of a transgene driven by 35S constitutive promoter that is anticipated to express the protein in tissues and at developmental stages where it is not expressed normally.

-Why is it stated in lines 189-191 that the two D class genes may have similar function when the KO lines point out they do not? It is stated explicitly in the discussion that they have different functions. This text should be clarified

Answer: It is true that the knockout of the two genes leads to completely divergent phenotypes which suggest divergent functionalities. Nevertheless, the phenotypic differences between the *SIMBP3*-KO and *SIAGL11*-KO lines do not necessarily mean

that the two proteins are functionally divergent. The ability of both genes to compensate the All-flesh phenotype and the similarity of the phenotypes of their OX lines suggest that are capable to have similar functions. The two genes are expressed in different territories except in seeds where GUS reporter driven by their promoter shows convergent expression in seed structure. It cannot be ruled out that *SIMBP3* may compensate for *SIAGL11* mutation which explain the absence of visible phenotypes in *SIAGL11*-KO lines. Altogether, these data support the notion that the two class D tomato proteins may share some functionalities.

REVIEWER COMMENTS

Reviewer #1 (Remarks to the Author):

The authors revised manuscript satisfactorily and I learned that the constitutive expression of SIAGL11 in a SIMBP3-KO mutant restored locular development, which strongly supports the authors findings, the functional redundancy of two homologues and the cause of the difference between an AC-RNAi fruit and a KO-fruit. I satisfied the revised manuscript and have no more comments or suggestions.

Reviewer #2 (Remarks to the Author):

The authors have convincingly responded to my criticisms and suggestions. By adding new experiments showing the histological difference in fruit development between WT and SIMB3-KO, they have supported the novel finding that the process determining fruit firmness is regulated at a very early stage of fruit development. They also provide new insights into unraveling the role of SIMB3 in the transcriptional reprogramming underlying the formation of locular gel tissue.

Reviewer #3 (Remarks to the Author):

I am grateful for the authors response to my prior review. The inclusion of the new data describing the effects of MBP3 on the progression of locule development helped clarify many prior concerns and the overall presentation. I have no further comments.